# A New AI-Based Semantic Cyber Intelligence Agent

Fahim Sufi 

School of Public Health and Preventive Medicine, Monash University, Melbourne, VIC 3004, Australia;
fahim.sufi@monash.edu

**Abstract:** The surge in cybercrime has emerged as a pressing concern in contemporary society due to its far-reaching financial, social, and psychological repercussions on individuals. Beyond inflicting monetary losses, cyber-attacks exert adverse effects on the social fabric and psychological well-being of the affected individuals. In order to mitigate the deleterious consequences of cyber threats, adoption of an intelligent agent-based solution to enhance the speed and comprehensiveness of cyber intelligence is advocated. In this paper, a novel cyber intelligence solution is proposed, employing four semantic agents that interact autonomously to acquire crucial cyber intelligence pertaining to any given country. The solution leverages a combination of techniques, including a convolutional neural network (CNN), sentiment analysis, exponential smoothing, latent Dirichlet allocation (LDA), term frequency-inverse document frequency (TF-IDF), Porter stemming, and others, to analyse data from both social media and web sources. The proposed method underwent evaluation from 13 October 2022 to 6 April 2023, utilizing a dataset comprising 37,386 tweets generated by 30,706 users across 54 languages. To address non-English content, a total of 8199 HTTP requests were made to facilitate translation. Additionally, the system processed 238,220 cyber threat data from the web. Within a remarkably brief duration of 6 s, the system autonomously generated a comprehensive cyber intelligence report encompassing 7 critical dimensions of cyber intelligence for countries such as Russia, Ukraine, China, Iran, India, and Australia.

**Keywords:** cyber intelligence; cyber threat analysis; cyber war; situational analysis; semantic agents; multi-agent communication

## 1. Introduction

By the year 2020, the global cost incurred as a result of cyber-attacks had surpassed $1 trillion USD [1]. Projections indicate that by 2025, cybercrimes will have an annual cost of up to $10.5 trillion USD [2]. Notably, larger economies, including China, Brazil, the United States, India, Mexico, France, Australia, and the United Arab Emirates, face billions of dollars in consumer losses due to cybercrime [3]. For instance, in 2017, Chinese consumers experienced a financial loss of $66.3 billion USD attributed to cybercrime [3]. Moreover, apart from financial ramifications, cyber-attacks have profound social and psychological implications for individuals [4]. A recent example is the cyber-attack on Optus, a prominent Australian telecommunications company, which resulted in widespread stress and anger among the affected individuals [5,6].

As a consequence of this cyber-attack, sensitive information, such as names, dates of birth, email addresses, driver's license details, Medicare cards, passport numbers, and more, may have been exposed, impacting 2.1 million people [6,7]. More recently, cybercriminals managed to obtain confidential patient data from Australia's leading health insurance provider, which included medical diagnoses and procedures [8]. Thus, cybercrime stands as a paramount challenge for contemporary countries, states, corporations, and individuals.

To mitigate the effects of cybercrime, two crucial prerequisites are the availability of cyberspace-related data and advanced analytical algorithms capable of detecting and preventing threats. Recent studies [1,9] have emphasized the importance of cyber data.

Such data can be acquired through actual network traffic analysis [10–13], simulation methods [14], surveys [15], open-source antivirus intelligence [16], and even social media data [17–23]. Among these various sources of cyber intelligence, open-source data and social media data from platforms such as Twitter have been identified as the most effective means of obtaining cyber data, as supported by recent research findings [16–23]. This paper introduces a novel multi-agent system designed to achieve the generation of multi-dimensional cyber intelligence through semantic communication among its constituent agents. The proposed agent effectively utilizes both social media data and web-based data to deliver a rigorous and comprehensive intelligence output to cyber analysts and strategists. Notably, the generation of cyber intelligence capitalizes on cutting-edge advancements in artificial intelligence (AI) and machine learning (ML) algorithms.

Figure 1 illustrates the schematic representation of the system wherein a user submits a request for cyber intelligence acquisition to the aggregation agent. The aggregation agent facilitates communication with two distinct agents, namely, the social media agent and the web media agent. The social media agent leverages various algorithms, including sentiment analysis, translation, term frequency analysis, topic modelling, and anomaly detection, to deliver cyber intelligence based on social media platforms, such as Twitter. These algorithms are implemented through both application programming interfaces (APIs) and non-API methods. The execution of API-based natural language processing (NLP) algorithms, such as sentiment analysis and translations, is carried out by a separate agent called the cognitive service agent.

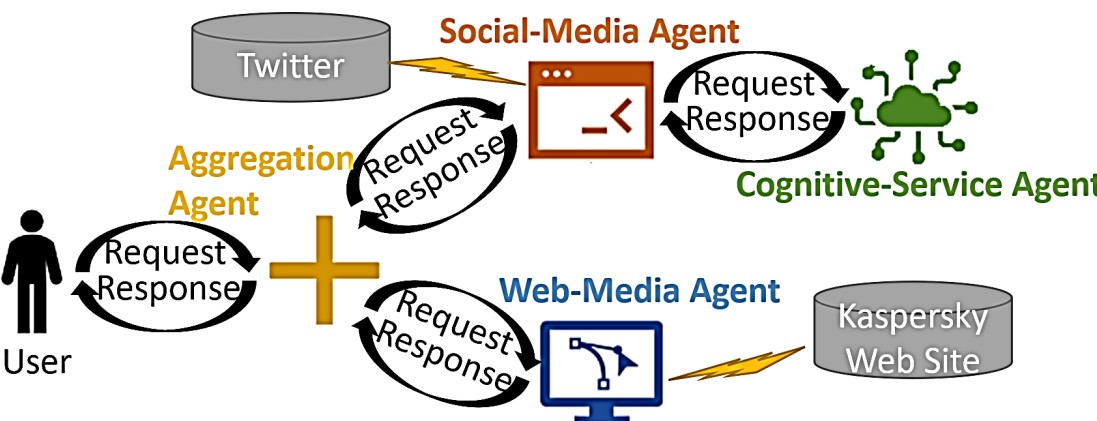

**Figure 1.** Schematic Diagram of Multi-Agent Cyber Intelligence System.

In addition to acquiring cyber intelligence from social media sources, the web media agent also provides cyber threat intelligence by encompassing the spectrum of threats specific to selected countries, which is sourced from antivirus vendor websites, such as Kaspersky. Finally, the user is furnished with a comprehensive cyber intelligence report that amalgamates knowledge obtained through AI-driven acquisition from both social media and web media sources.

Table 1 presents a comprehensive overview of the distinctive features of the four unique designs proposed in this paper along with their corresponding meaningful outputs. Of particular significance is the semantic output generated for the user, encompassing various elements, such as country name, threat level, threat spectrum, geopolitical and socioeconomic factors, physiological and societal aspects, impacted targets, national concerns, and victimization. This comprehensive cyber intelligence output aligns with the prevailing literature on the subject matter [24–30].

**Table 1.** Features and Semantic Output of Cyber Intelligence Agents.

| Name of Agent | Feature | Semantic Output |
|---|---|---|
| Aggregation Agent | (1) Interact with User via Web, iOS, and Android Devices<br>(2) Collaborate with Social Media Agent<br>(3) Collaborate with Web Media Agent<br>(4) Generate Multi-Dimensional and Multi-Source Comprehensive Cyber Intelligence on Selected Countries | (1) Country Name:<br>(2) Threat Level:<br>(3) Threat Spectrum:<br>(4) Geopolitical/ Socioeconomic:<br>(5) Psychological and Societal:<br>(6) Impacted Target:<br>(7) National Concern:<br>(8) Victimization: |
| Social Media Agent | (1) Obtain Social Media Data<br>(2) Collaborate with Aggregation Agent<br>(3) Collaborate with Cognitive Service Agent<br>(4) Generate Term Frequency<br>(5) Generate Topic Modelling<br>(6) Deep Learning-Based Anomaly Detection | (1) Country Name:<br>(2) Word Frequency:<br>(3) Topics with Word Frequencies:<br>(4) Sentiments on Time-Series:<br>(5) Alerts on Time-Series:<br>(6) Anomalies on Time-Series: |
| Cognitive Service Agent | (1) Collaborate with Social Media Agent<br>(2) Generate Translation<br>(3) Generate Sentiment Analysis | (1) Original Language:<br>(2) Translated Text:<br>(3) Overall Sentiment:<br>(4) Sentiment Confidence: |
| Web Media Agent | (1) Obtain Cyber Threat Statistics on Malicious Mail, Ransomware, Exploits, Web Threats, Spam, Local Infection, Network Attacks, On-Demand Scans from Web Data<br>(2) Collaborate with Aggregation Agent<br>(3) Generate Multi-Dimensional Threat Spectrum<br>(4) Deep Learning-Based Anomaly Detection<br>(5) Threat Prediction | (1) Country Name:<br>(2) Threat Type:<br>(3) Country Rank:<br>(4) Threat Percentage:<br>(5) Anomalies on Time-Series:<br>(6) Threat Prediction on Time-Series: |

In order to assess the efficacy of the newly devised social media-based cyber threat intelligence system, a comprehensive analysis was conducted on a dataset comprising 37,386 Tweets authored by 30,706 unique users. The data collection period spanned from 13 October 2022 to 6 April 2023. Throughout this timeframe, a diverse range of 54 languages was captured, processed, and subjected to in-depth analysis. The resulting outcome yielded multi-dimensional cyber threat intelligence specifically tailored for the countries of China, Russia, Ukraine, Australia, Iran, and India.

## 2. Background and Literature Review

This section lays the foundations for multi-dimensional cyber intelligence as well as existing literature on NLP.

### 2.1. Contextual Information on Multi-Dimensional Cyber Intelligence

Ref. [26] presents a comprehensive framework for categorizing and evaluating the impacts of cyber-attacks on individuals, organizations, and society while also highlighting current limitations and challenges in cyber harm research and offering recommendations for future studies. The paper examines four case studies to establish connections between various forms of harm and the potential spread of cyber violence, emphasizing the need for analytical tools for organizational cyber harm based on a proposed taxonomy. Ref. [27] identifies and analyses common cyber security vulnerabilities and their recurrence rates, including the affected publication venues, countries, and targeted infrastructures and applications. The report emphasizes the necessity for further research in identifying critical vulnerabilities and developing effective mitigation strategies, emphasizing the need for empirical validation and practical implementation. Ref. [28] underscores the significance of employee and organizational education, training, and awareness in preventing and minimizing cyber security incidents, outlining major obstacles, and offering a research framework in this area. Ref. [29] focuses on phishing attacks, examining previous strategies, assessing the current landscape, and proposing a comprehensive anatomy of phishing

that covers attack phases, attacker types, weaknesses, threats, targets, attack media, and strategies with the aim of raising awareness and aiding the development of an anti-phishing system. Ref. [30] explores various perspectives on cyber security threats, including technical, human, organizational, and environmental factors, and proposes a methodology to examine these concerns and their interactions. The article suggests a multi-dimensional approach to identify underlying causes and develop more robust defences. Table 2 provides an overview of the research works (i.e., [24–30]) and the strategic questions they address along with the dimension of cyber threat.

**Table 2.** Strategic questions answered by multi-dimensional cyber threat intelligence.

| Strategic Questions Answered | Dimension of Cyber Threat | Reference |
|---|---|---|
| 1. What type of threat?<br>2. Who is attacking? | Threat spectrum (e.g., malware, spyware) | [24,27,29,30] |
| 3. Where is the attack coming from?<br>4. Why is the attack happening?<br>5. What is the motivation for this attack? | Geopolitical and socioeconomic | [30] |
| 6. Who is the target?<br>7. Who is the victim of the cyber-attack? | Victimization (human vs. system) | [24,25] |
| 8. What are the major cyber-related concerns? | National priority and concerns | [26,28] |
| 9. What is the impact? | Impacted target (infrastructure, supply chain, etc.) | [24] |
| 10. What is the societal perception?<br>11. How do cyber-attacks affect society?<br>12. How much negativity is generated at a psychological level? | Psychological and societal | [26] |
| 13. What is the severity level of the threat?<br>14. What is the intensity of the cyber threat? | Threat level (low, medium, high) | [24] |

### 2.2. NLP-Based Cyber Intelligence from Social Media

Analysing cyber-related social media posts on Twitter began nearly a decade ago [31], but these early studies did not leverage the potential of machine learning (ML) and deep learning (DL) techniques for automated critical analysis. Instead, they relied on the manual application of general statistical techniques to gain insights into cyberbullying and related issues. In [17], a methodology utilizing sentiment analysis of tweets is described for predicting cyber-attacks. The authors employed machine learning algorithms to categorize tweets as security-related, positive, or neutral and then examined the relationship between sentiment scores and cyber-attacks reported by Google News using a regularised regression model. Their findings suggest that this methodology can serve as a warning system for identifying potential cyber-attacks, particularly when the coefficient of determination is high. They also discuss the response of hacktivists to candidates' comments and actions in the 2016 US presidential elections, providing examples. Ref. [21] utilizes the term frequency-inverse document frequency (TF-IDF) to extract features from a dataset of 2000 tweets, comparing the performance of five classifiers and determining that logistic regression is the best classifier for detecting bullying tweets. In [22], TF-IDF and LGBM algorithms are employed to identify cyber-attacks in darknet traffic, achieving a high accuracy of 98.97% compared to other algorithms. Ref. [19] employs two machine learning-based classifiers to analyse a large-scale Twitter dataset, revealing negative sentiment across various themes. Ref. [20] identifies sensitive keywords that could lead to vulnerability when combined with benchmarked cyber keywords using LDA, while [23] analyses COVID-19 misinformation spread on Twitter through sentiment, emotion, topic, and user attributes, highlighting the denial of the pandemic and dissemination of false information. Table 3 showcases the diverse NLP algorithms used in these studies, with the proposed system presenting the most comprehensive use of NLP-based algorithms for generating seven-dimensional cyber intelligence.

**Table 3.** Existing research on analysing cyber-related social media posts using NLP algorithms (X denotes Supported).

| Reference | Sentiment Analysis | Translation | LDA | TF-IDF | Stemming | N-Gram | Forecasting | ML Algorithms |
|---|---|---|---|---|---|---|---|---|
| [17] | X | | | X | X | X | X Regression | Naïve Bayes Classifier, Support Vector Machines, Maximum Entropy Classifier |
| [18] | X | X | | | | | | |
| [19] | X | | X | X | | X | | BERT-based, Logistic Regression, SVM, Random Forest, XGBoost |
| [20] | | | X | X | | X (bi-Gram) | | |
| [21] | | | | X | | | | Support Vector Classifier, Logistic Regression, Naïve Bayes, Random Forest Classier, SGD Classifier |
| [22] | | | | X | | | | LightGBM (light gradient boosted machine) |
| [23] | X | | X | | | | | |
| Proposed | X | X | X | X | X | X | X | CNN (Deep Learning) |

## 3. Materials and Methods

The proposed methodology incorporates the development of four distinct agents that interact with each other in a complex manner to fulfil the user's goal of acquiring AI-based cyber intelligence across various countries. Figure 2 illustrates the integrated semantic multi-agent communication within this framework. Figures 2–4 illustrate the building blocks and detailed sub-processes of the four agents. These sub-processes (i.e., language detection and translation, sentiment analysis, anomaly detection, term frequency generation, topic generation, and threat prediction) are explored here in further detail within Section 3.1, Section 3.2, Section 3.3, Section 3.4, Section 3.5, Section 3.6.

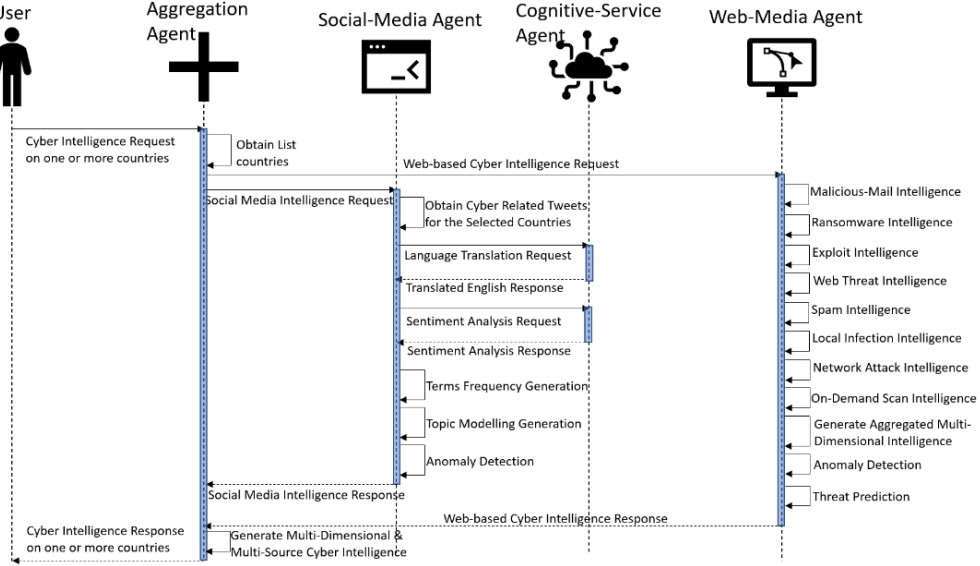

**Figure 2.** UML sequence diagram of multi-agent communications with user.

As seen in Figure 3, social media agent and cognitive service agent perform several NLP activities (e.g., translation, sentiment analysis, term frequency generation, and topic generation) and deep learning activity (i.e., anomaly detection) on Twitter data [16]. On the other hand, web media agent performs deep learning tasks, such as anomaly detection, and statistical modelling tasks, such as threat prediction, from web media data on cyber-attacks [18], as seen in Figure 4. Hence, the proposed architecture of multi-agent communication in semantic manner realizes the first multi-source cyber intelligence with NLP and AI via the amalgamation of both social media [16] and web data [18].

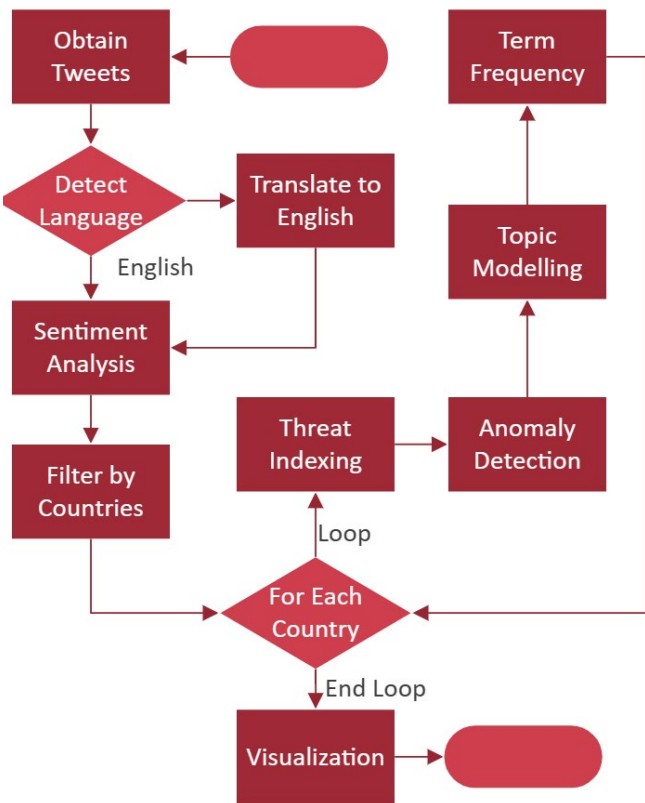

**Figure 3.** Flowchart of the activities of social media agent and cognitive service agent.

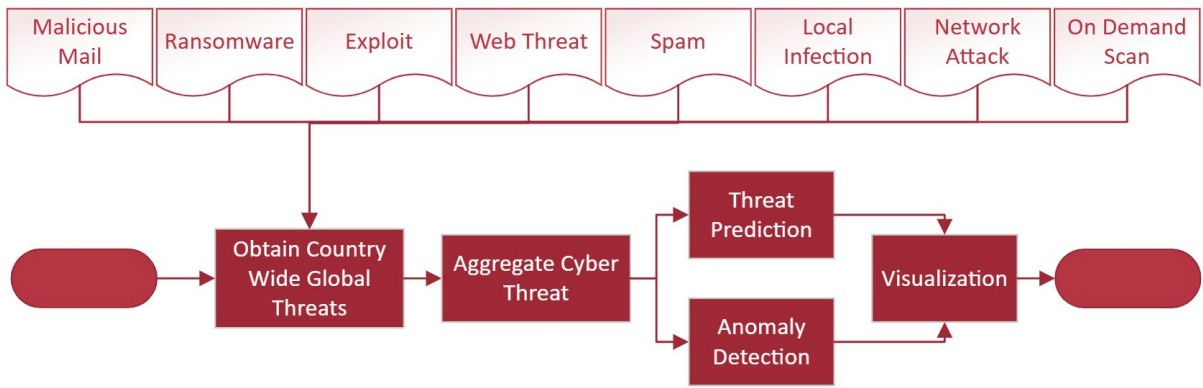

**Figure 4.** Flowchart of activities of web media agent.

### 3.1. Language Detection and Translation Process

Microsoft's Azure Cognitive Services empowers developers to build intelligent applications without the requirement of developing the underlying AI technology. It offers language translation and detection functionalities. Language detection is the process of identifying the language of a given text, and Azure Cognitive Services provides a Language Detection API capable of automatically determining the language of text inputs. This API supports various programming languages, such as C#, Java, Python, and more, and demonstrates remarkable accuracy in detecting over 120 different languages [32]. As seen from Algorithm 1 (line 1 to line 5), this process converts a multilingual stream of social media messages $x_i$ to $y_i$, where $i = 0$ to $N$. Appendix A.3 provides sample codes on the detailed usage of language detection and translation process.

---

**Algorithm 1:** Language Processing on Cyber-Related Social Media Messages

---

1: *For each $x_i$ in N, Multilingual Social Media Messages*
2:     *If Language($x_i$)<> 'English'*
3:         *$y_i$ = Translate($x_i$)*
4:     *Else*
5:         *$y_i$ = $x_i$*
6: *For each $y_i$ in N, English Social Media Messages*
7:     *$s_i$ = Sentiment($y_i$)*
8:     *If $y_i$ Contains 'Country Name'*
9:         *$\{c_r, y_i, t_{c_r}\}$= $y_i$*
10: *For each $c_r$ in C, Countries*
11:     *$\{Yes/No, t_r\}$ = AnomalyDetection(CountofMessagesonTimeUnit($t_r$),$t_r$)*
12:     *$\{\{w_j, f_{w_j}\}, \dots \}$ = TermFrequency(Tokenize($y_i$))*
13:     *$\{\{w_k, f_{w_k}\}, \dots \}$ = Stemming(Tokenize($y_i$))*
14:     *$\{\{\bigcup_1^l w_l, f_o\}, \dots \}$ = n_gram(Tokenize($y_i$))*
15:     *$\{\{v_p, \{\{w_q, f_{w_q}\} \dots \}\}, \dots \}$ = Topic(Tokenize($y_i$))*
16: *Generate Interactive Visualization*

---

### 3.2. Sentiment Analysis Process

Sentiment analysis, the process of determining the emotional tone conveyed in a text, can be effectively performed using the Sentiment Analysis API provided by Azure Cognitive Services [32]. This API utilizes natural language processing techniques to analyse sentiments associated with phrases, entities, and linguistic constructions present in the input text. With a machine learning model trained on extensive text data, the API offers highly accurate sentiment analysis for various text inputs, including social media posts, customer reviews, and support issues. Users can access the API by registering for an Azure account and subscribing to the Sentiment Analysis API and integrating it into their applications through SDKs or REST APIs available for multiple programming languages. Alongside an overall sentiment score, the API provides detailed information, such as sentiment scores for individual sentences and key phrases/entities associated with each sentiment score. This information can be utilized in customer service, marketing, and social media monitoring to make informed decisions and gain deeper insights into expressed sentiments [17,19,33]. Appendix A.3 provides sample codes on the detailed usage of sentiment analysis process.

$$s_i = Sentiment(y_i) \tag{1}$$

$$Sentiment\ Classification = \begin{cases} Positive, if\ s_i \geq 0.7 \\ Negative, if\ s_i \leq 0.3 \\ Neutral, if\ 0.3 > s_i > 0.7 \end{cases} \tag{2}$$

### 3.3. Anomaly Detection Process

The anomaly detector enhances line charts by automatically identifying anomalies in time-series data. It employs NLP-based root cause analysis to dynamically explain the detected anomalies [32–34]. In this section, the problem definition is discussed first. When a sequence of real values is presented, $x = x_1, x_2, x_3, \dots, x_n$, time-series anomaly detection's target becomes producing an output sequence of $y = y_1, y_2, y, \dots, y_n$, where $y_i \in \{0, 1\}$ denotes whether $x_i$ is an anomaly point.

Research in [35] demonstrated the process of saliency reduction (SR) from visual saliency detection domain followed by application of CNN to the output of SR model. This study implements similar process as prescribed in [35] with the following three core tasks:

Apply Fourier transform for generating log amplitude spectrum.

Compute the SR.

Apply inverse Fourier transform for transforming the sequence back to the spatial domain.

$$A(f) = Amplitude(f(x)) \tag{3}$$

$$P(f) = Phrase(f(x)) \tag{4}$$

$$L(f) = log(A(f)) \tag{5}$$

$$AL(f) = h_q(f) \times L(f) \tag{6}$$

$$R(f) = L(f) - AL(f) \tag{7}$$

$$S(x) = \left| \left| f^{-1}(exp(R(f) + iP(f))) \right| \right| \tag{8}$$

Here, Fourier transform and inverse Fourier transform are represented by $f$ and $f1$, respectively. Moreover, $x$ represents the input sequence with shape $nX1$, and $A(f)$ represents amplitude spectrum of sequence $x$. Furthermore, phase spectrum of sequence $x$ is denoted by $P(f)$. Log representation of $A(f)$ is represented here with $L(f)$; then, average spectrum of $L(f)$ is presented with $AL(f)$, which can be estimated by convoluting the input sequence by $hq(f)$. Here, $hq(f)$ can be presented with a $q \times q$ matrix as shown in Equation (9).

$$h_q(f) = \frac{1}{q^2} \begin{bmatrix} 1 & 1 & \dots 1 \\ 1 & 1 & \dots 1 \\ \dots & \vdots & \ddots 1 \\ 1 & 1 \dots & 1 \end{bmatrix} \tag{9}$$

As shown in Equation (7), $R(f)$ is calculated by subtracting the averaged log spectrum $AL(f)$ from the log spectrum $L(f)$. Here, SR is denoted with $R(f)$. Finally, as shown in Equation (8), by applying an inverse Fourier transform, the sequence was assigned back to the spatial domain. The final output sequence $S(x)$ represented within Equation (9) is called the saliency map [36]. The anomaly points are computed with Equation (10).

$$x = (\overline{x} + mean)(1 + var) \times r + x \tag{10}$$

Within Equation (10), the local average of the preceding points is represented by $\overline{x}$. On the other hand, within Equation (10), mean and var are the mean and variance of all points within the current sliding window (i.e., randomly sampled $r \sim N(0, 1)$). In this manner, CNN is employed on the saliency map (i.e., not on the raw input). The procedure of anomaly detection thereby maximizes efficacy and efficiency [35,36]. As previously indicated, the anomaly identification method used in this study makes use of NLP to provide plain English explanations of the origins of all the anomalies [32].

### 3.4. Term Frequency Generation Process

The text analysis method known as term frequency-inverse document frequency (TF-IDF) is used to determine the relative value of words in a document [17,19–22]. A word's frequency in a document is multiplied by the word's inverse frequency over the entire corpus to arrive at this number (inverse document frequency). The resulting score gives terms that are significant in a specific document but uncommon across the entire corpus greater weight. As seen from line 12 of Algorithm 1, the TF-IDF process generates a vector of words along with frequencies of each word as $\{\{w_j, f_{w_j}\}, \dots \}$. Here, $w_j$ is a word, and $f_{w_j}$ is the corresponding frequency.

Porter stemming is a method for condensing words to their root or fundamental form [17]. It normalizes words by deleting frequent suffixes from them. For instance, the roots of the words "running", "runs", and "run" would all be the same. By lowering the number of unique terms and combining similar words, this method can increase the accuracy of text analysis. Line 13 of Algorithm 1 demonstrates the creation of vector $\{\{w_k, f_{w_k}\}, \dots \}$, where $w_k$ is the root word and $f_{w_k}$ is the corresponding frequency.

In a text, n-grams are continuous groups of n words [17,19,20]. They are frequently employed in text analysis to record linguistic context and structure. For instance, trigrams ($n = 3$) can capture more complicated word sequences, such as "deep learning algorithms", while bigrams ($n = 2$) can capture pairs of words that frequently occur together, such

as "machine learning". For text classification, clustering, and other text analysis tasks, n-grams can be utilized to produce features. Line 14 of Algorithm 1 shows unions of words (i.e., $\bigcup_1^l w_l$) and their corresponding frequency, $f_o$. For bigrams, it is $w_1 \cup w_2$, and for trigrams, it is $w_1 \cup w_2 \cup w_3$. TF-IDF, Porter stemming, and n-grams can all be used in conjunction to preprocess and analyse text data, which can enhance the precision and efficiency of text analysis operations.

### 3.5. Topic Generation Process

A method for locating the key themes or subjects in a group of texts is called topic modelling. Latent Dirichlet allocation (LDA) is one of the most well-liked topic modelling algorithms [19,20]. According to the LDA statistical model, each document in a collection is composed of a few different themes, and each word is produced by one of those topics. In order for the LDA algorithm to function, each word in each document is first given a topic at random. The subject assignments are then incrementally updated based on the words found in the texts until a set of topic assignments is found that most effectively explains the words found. The LDA method can be used to study and investigate the content of the documents in a variety of ways after it has determined the subjects in the document collection [19,20]. It can be used, for instance, to find the subjects that appear most frequently in the collection, to investigate the relationships between the topics, and to find the documents that are most closely related to each topic. LDA has a wide range of applications in areas including natural language processing, text mining, and information retrieval and is a potent tool for revealing latent subjects in a collection of texts. As shown in line 15 of Algorithm 1, topics $v_p$ and a list of words along with corresponding frequencies of the topic $\{w_q, f_{w_q}\}$ are presented by the LDA algorithm.

### 3.6. Threat Prediction Process

The study in [37] utilizes exponential smoothing, a statistical method, to predict malware attack frequency. It employs mean squared error (MSE), mean absolute error (MAE), and mean absolute percentage error (MAPE) to evaluate single and double exponential smoothing models. The results indicate that single exponential smoothing is effective in predicting malware attacks, and therefore this technique is also adopted here. The only smoothing parameter used in this single exponential model is $\alpha$. The fundamental premise guiding this single exponential smoothing is that the data's mean is steady while the actual data fluctuates around it. Thus, a single exponential smoothing model is represented in Equation (11). The double exponential smoothing is an expansion of the single exponential smoothing. The double exponential smoothing method uses two parameter constants, $\alpha$ and $\beta$. The smoothing constant $\alpha$ is used to smooth the level value estimate, and $\beta$ is used to smooth the trend value estimate. Double exponential smoothing is employed when there is evidence of a trend in the data but no seasonality effect. It is similar to single exponential smoothing, but $\alpha$ and $\beta$, which are the level and trend smoothing parameters, must be changed every time. Equation (11) represents double exponential smoothing, Equation (12) shows the trend estimate, and Equation (13) represents forecast in m step ahead.

$$F_{t+1} = \alpha y_t + (1 - \alpha) F_t \tag{11}$$

$$L_t = \alpha y_t + (1 - \alpha)(L_{t-1} + b_{t-1}) \tag{12}$$

$$b_t = \beta(L_t - L_{t-1}) + (1 - \beta)b_{t-1} \tag{13}$$

$$F_{t+m} = L_t + mb_t \tag{14}$$

Here, $F_{t+1}$ (in Equation (11)) is the forecast for the next period, and $F_t$ is the old forecast for period $t$. On the other hand, $y_t$ is the actual value at $t$. Within Equation (12) $L_t$ is the estimate for the level of the time series at time $t\alpha$ = smoothing constant for the data, $b$ is

smoothing constant for trend estimate, $b_t$ is the estimate of the slope of the series at time $t$, and $m$ represents the periods to be forecast into the future.

Earlier in this section, Figures 2–4 presented the building blocks of the prosed systems. These building blocks are subsequently described and summarized in Table 4. Each of these building blocks presented within this section was implemented with Algorithm 1.

**Table 4.** Algorithms used within the proposed AI-based cyber threat intelligence.

| Process Name | Algorithm Used | Algorithm Type | API Used | References |
|---|---|---|---|---|
| Sentiment Analysis | Microsoft Text Analytics | NLP | Yes | [17,19,33] |
| Translate to English | Microsoft Text Analytics | NLP | Yes | [33] |
| Anomaly Detection | CNN | Deep Learning | No | [32,33] |
| Topic Modelling | LDA | NLP | No | [19,20] |
| Term Frequency | TF-IDF | NLP | No | [17,19–22] |
| Term Frequency | Porter Stemming | NLP | No | [17] |
| Term Frequency | N-Gram | NLP | No | [17,19,20] |
| Forecast Threat | Exponential Smoothing | NLP | No | [37] |

## 4. Results

The proposed method was evaluated from 13 October 2022 to 6 April 2023 using a dataset of 37,386 tweets from 30,706 users in 54 languages. These tweets were obtained using Microsoft Power Automate as demonstrated in Figure 5. Acquiring tweets through Power Automate involves leveraging its capabilities to integrate with external platforms, such as Twitter's Application Programming Interface (API), as demonstrated in our research [16,33,34]. Power Automate, a cloud-based service provided by Microsoft, allows users to automate workflows and create custom applications without requiring extensive coding knowledge. To obtain tweets, one must first authenticate their Power Automate connection with the Twitter API by generating API credentials and obtaining an access token. This process involves registering a Twitter developer account, creating an application, and obtaining the necessary keys and tokens. Once authenticated, users can utilize Power Automate's "HTTP" action to send requests to the Twitter API's endpoints, such as the "search/tweets" endpoint. By specifying relevant parameters, such as search keywords, date range, or user handles, users can retrieve specific tweets or perform comprehensive searches. Power Automate can then process and manipulate the received tweet data, enabling users to store them in a database, send notifications, perform sentiment analysis, or integrate them into other applications, thereby empowering users to streamline their workflows and leverage Twitter's vast data resources. It should be mentioned that keywords such as "Cyber" and "Hack" were used to obtain the real-time tweets from 13 October 2022 to 6 April 2023. All tweets (regardless of their relevancy) were analysed in the proposed system.

A total of 8199 HTTP requests were made to translate non-English tweets. These data were processed by a social media agent. Apart from the social media data, 238,220 cyber threat data were obtained and processed by the web media agent during the same timeframe (13 October 2022 to 6 April 2023). As seen from Table 5, the web media agent collected about 30K attack statistics for each of the eight different attack types (e.g., exploits, local infections, malicious mail, network attacks, on-demand scans, ransomware, spam, and web threats). Kaspersky is a trusted provider of cyber threat statistics and follows multi-dimensional cyber-attack data obtained by web media agents with web scraping techniques:

- Daily ransomware data from https://statistics.securelist.com/ransomware/day (accessed on 3 March 2023)
- Daily vulnerability data from https://statistics.securelist.com/vulnerability-scan/day (accessed on 3 March 2023)
- Daily web threat data from https://statistics.securelist.com/web-anti-virus/day (accessed on 3 March 2023)

- Daily spam data from https://statistics.securelist.com/kaspersky-anti-spam/day (accessed on 3 March 2023)
- Daily malicious mail data from https://statistics.securelist.com/mail-anti-virus/day (accessed on 3 March 2023)
- Daily network attack data from https://statistics.securelist.com/intrusion-detection-scan/day (accessed on 3 March 2023)
- Daily local infection data from https://statistics.securelist.com/on-access-scan/day (accessed on 3 March 2023)
- Daily on-demand-scan data from https://statistics.securelist.com/on-demand-scan/day (accessed on 3 March 2023)

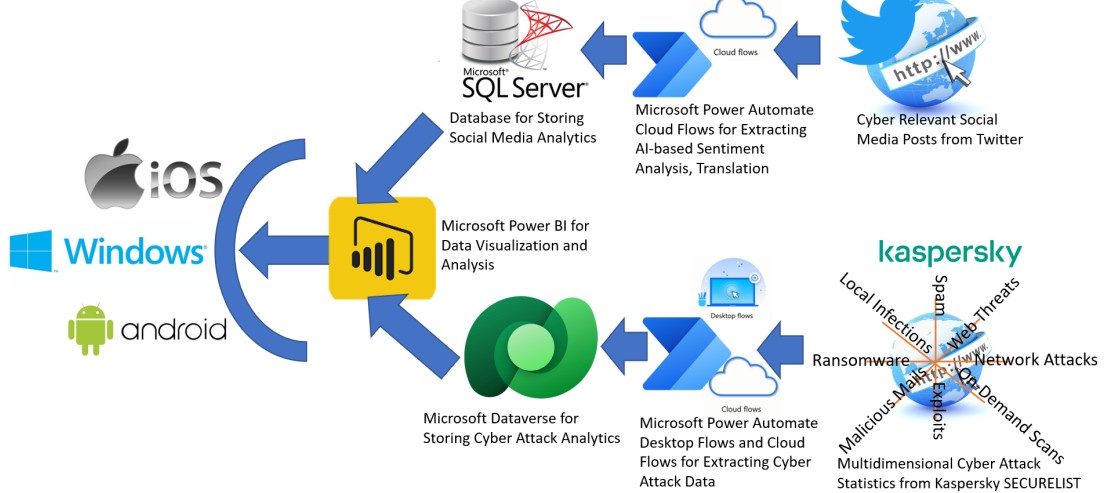

**Figure 5.** Deployment structure of the 4 cyber intelligence agents.

**Table 5.** Distribution of 238,220 cyber threat data by attack type.

| Attack Type | Exploit | Local Infection | Malicious Mail | Network Attack | On-Demand Scan | Ransomware | Spam | Web Threat |
|---|---|---|---|---|---|---|---|---|
| Number of Records | 29,017 | 32,592 | 30,165 | 30,522 | 32,584 | 23,299 | 27,450 | 32,591 |

It should be highlighted that all these multi-dimensional cyber threat statistics from Kaspersky's attack statistics (i.e., https://statistics.securelist.com/) (accessed on 3 March 2023) site provided a daily dump of threat statistics. The cloud-based web-media agent built with Microsoft Power Automate downloaded these statistics on a daily schedule and saved these statistics within Microsoft Dataverse, as demonstrated in Figure 5. The low-code implementation of the technique is also demonstrated in [18].

During the evaluation, the deployment structure portrayed in Figure 5 was used. Could-based Microsoft Power Platform and Microsoft Azure ecosystem were used for deploying the agents. Hence, industry-standard configurations were adopted for agent deployment. Agents in Microsoft Azure are deployed through Azure Automation, a cloud-based service facilitating the automation and orchestration of tasks across Azure resources and external systems. To initiate agent deployment, an Azure Automation account is created as the central management hub. Subsequently, the Azure Automation agent is installed on the target machine or virtual machine, enabling the execution of automation runbooks and establishing secure communication between the agent and the automation service. Configuration of agent settings, such as defining runbook worker groups, proxy settings, network access control rules, and resource management, follows the agent's successful connection to the Azure Automation account. By assigning runbooks to the deployed agent within the Azure Automation account, desired automation tasks or workflows can be executed on the target machine. This deployment process empowers

users to streamline their Azure resources through effective task automation and orchestration. As seen from Figure 5, technology building blocks, such as Microsoft SQL Server, Microsoft Dataverse, Microsoft Power Automate, and Microsoft Power BI, were used to generate AI-based cyber intelligence from both Twitter data and Kaspersky's attack statistics (i.e., https://statistics.securelist.com/) (accessed on 3 March 2023).

Table 6 presents comprehensive information on the Twitter data collected over a 7-month period. It includes the time frame, number of tweets, unique users, unique locations, unique languages, total retweets, average confidence scores for negative, neutral, and positive sentiment analyses, and the number of translated tweets. The table reveals an increase in the quantity of tweets and users over time, while the number of languages remained stable. Retweet counts varied each month, with November 2022 having the highest total. The average confidence scores for sentiment analyses remained stable, with negative sentiment showing higher confidence than neutral and positive sentiment. Only a small portion of tweets were translated, with December 2022 having the highest percentage. Figure 6 displays average daily sentiment score fluctuations on time-series data as produced by the cognitive service agent. Figure 7 focuses solely on average monthly negative sentiments for Russia, China, Australia, Ukraine, India, and Iran. As highlighted in Figure 7, the monthly average of negative sentiment for Russia was at its peak in January 2023 with an average negative sentiment confidence of 0.74. Negative sentiment cyber-related posts are considered alerts [16]. These alerts or negative sentiments produced by the cognitive service agent were used to generate the psychological effect (i.e., number 5 semantic output of the aggregation agent shown in Table 1) of the cyber threat dimension.

**Table 6.** Processing of Tweets for AI-based Cyber-Threat Intelligence.

| Time | No. of Twitters | No. of Users | No. of Locations | No. of Languages | Total Retweets | Avg. Confidence of − Ve Seti. | Avg. Confidence of Neut. Seti. | Avg. Confidence of + Ve Seti. | No. of Translations |
|---|---|---|---|---|---|---|---|---|---|
| October 2022 | 3954 | 3556 | 1588 | 38 | 3,727,756 | 0.36 | 0.43 | 0.21 | 941 |
| November 2022 | 6470 | 5875 | 2358 | 38 | 9,981,856 | 0.34 | 0.43 | 0.23 | 1283 |
| December 2022 | 6512 | 5544 | 2225 | 42 | 7,565,946 | 0.35 | 0.42 | 0.23 | 1533 |
| January 2023 | 6685 | 5785 | 2364 | 40 | 7,802,301 | 0.36 | 0.40 | 0.24 | 1419 |
| February 2023 | 5976 | 5053 | 2114 | 43 | 4,276,479 | 0.37 | 0.42 | 0.21 | 1373 |
| March 2023 | 6634 | 5749 | 2357 | 41 | 4,799,540 | 0.36 | 0.43 | 0.21 | 1469 |
| April 2023 | 1155 | 1083 | 538 | 27 | 713,083 | 0.40 | 0.41 | 0.20 | 258 |
| Total | 37,386 | 30,706 | 10,178 | 54 | 38,866,961 | 0.36 | 0.42 | 0.22 | 8199 |

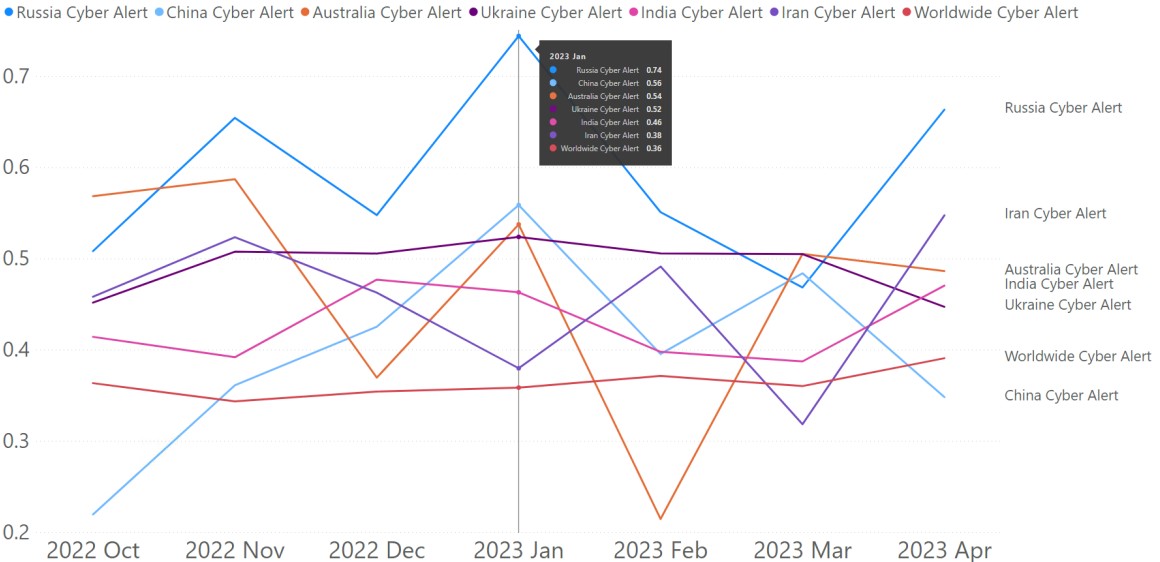

**Figure 6.** Daily average of tweet sentiments (8 November 2022 recorded the highest level of negative sentiment).

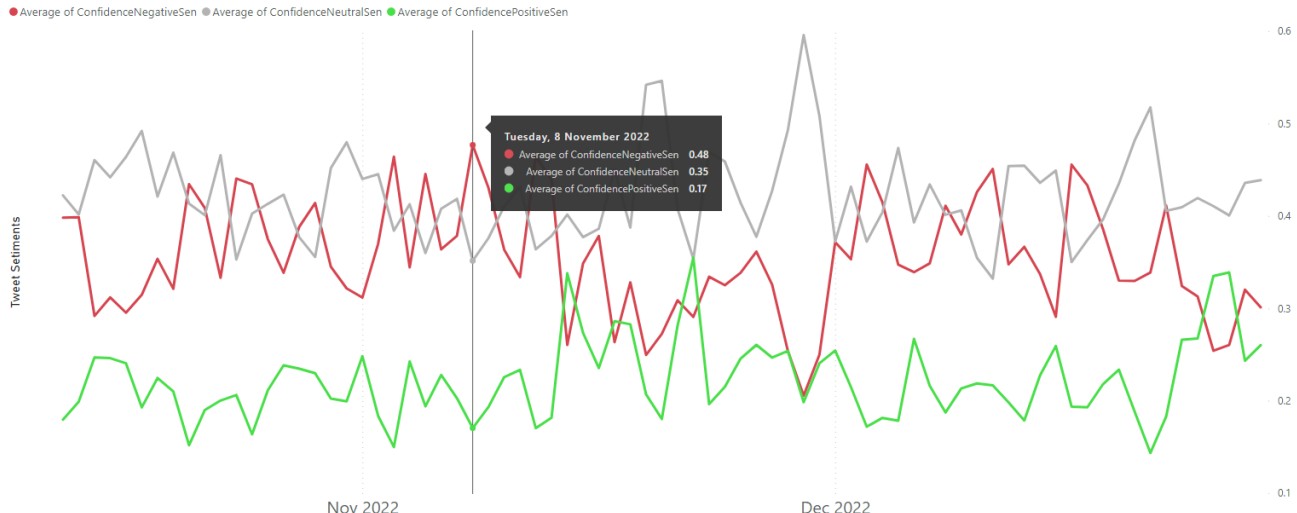

**Figure 7.** Monthly average of alerts (i.e., negative sentiments) for Russia, China, Australia, Ukraine, India, and Iran.

The CNN-based anomaly detection algorithm identified 7 anomalies for Russia, 5 anomalies for India, 5 anomalies for Iran, 4 anomalies for Ukraine, 4 anomalies for China, and finally, 12 anomalies for Australia, as depicted in Figure 8. These anomalies were identified by the social media agent by analysing live tweets during the monitored period. It should be mentioned that the number of tweets for each of the countries varied every month depending on the number of cyber-related issues and how the general public reacted psychologically to those issues. Figure 9 demonstrates the ranking of the number of tweets for each of the monitored countries between 13 October 2022 and 6 April 2023. As seen from Figure 9, in January 2023, the number of tweets for Russia was highest followed by Ukraine, India, Iran, China, and Australia. These statistics were slightly different in other months. However, cyber-related posts for Russia remained the focal point of discussion among Twitter users.

These tweets were analysed subsequently using term frequency (as shown in Table 7) and topic modelling (as shown in Tables 8 and 9).

**Table 7.** Result of Porter stemming for extracting 18 top-most frequent words of cyber tweets on China, Russia, Ukraine, India, and Australia.

|  | China | Russia | Ukraine | India | Australia |
|---|---|---|---|---|---|
| 1 | china | russian | ukrain | cyber | australian |
| 2 | cyber | russia | cyber | india | cyber |
| 3 | http | cyber | http | http | australia |
| 4 | hack | hack | hack | indian | http |
| 5 | russia | attack | russia | hack | secur |
| 6 | attack | http | russian | secur | hack |
| 7 | chines | trump | ukrainian | crime | polic |
| 8 | hacker | us | attack | attack | data |
| 9 | state | putin | militari | account | report |
| 10 | countri | stori | make | awar | attack |
| 11 | secur | timothydsnyd | secur | cybersecur | commun |
| 12 | backdoor | ukrain | year | govern | cybersecur |
| 13 | nation | heard | countri | polic | care |
| 14 | compani | sourc | help | pleas | media |
| 15 | access | afterward | defens | youtub | million |
| 16 | admin | april | forc | china | zealand |
| 17 | cybersecur | broke | invas | bank | accus |
| 18 | databas | intim | report | compani | custom |

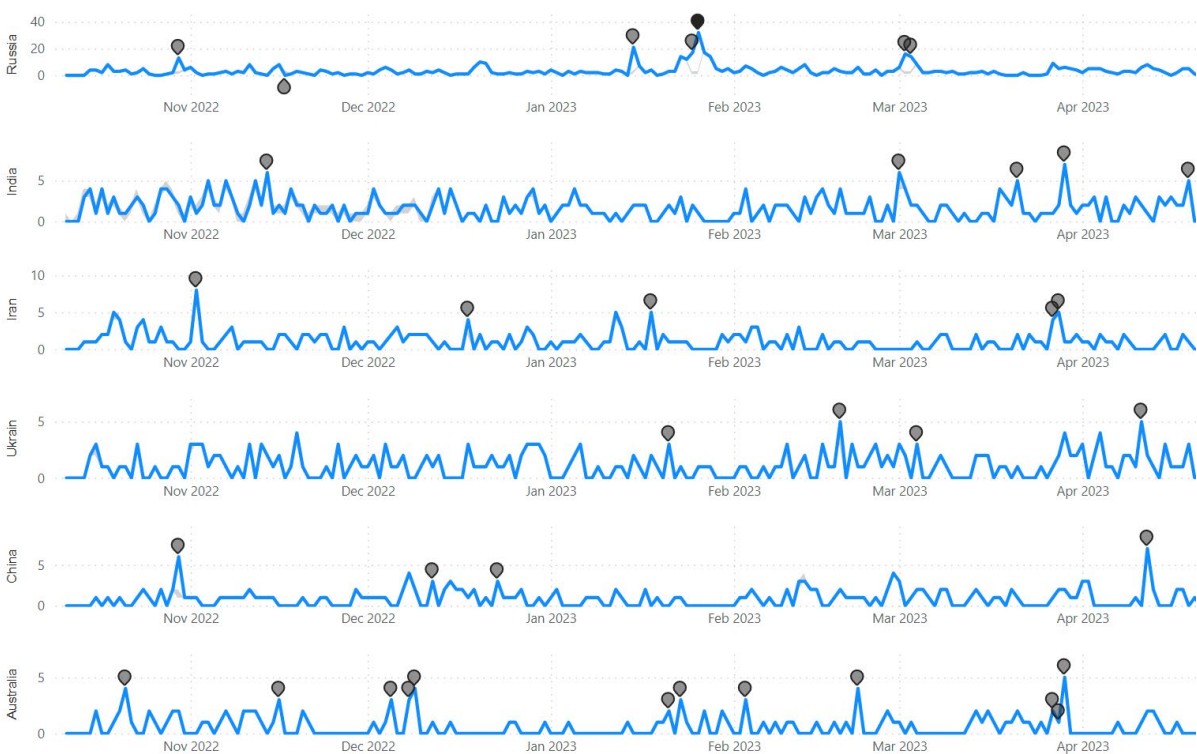

**Figure 8.** Anomaly detection detected cyber anomalies in Russia, China, Australia, Ukraine, India, and Iran.

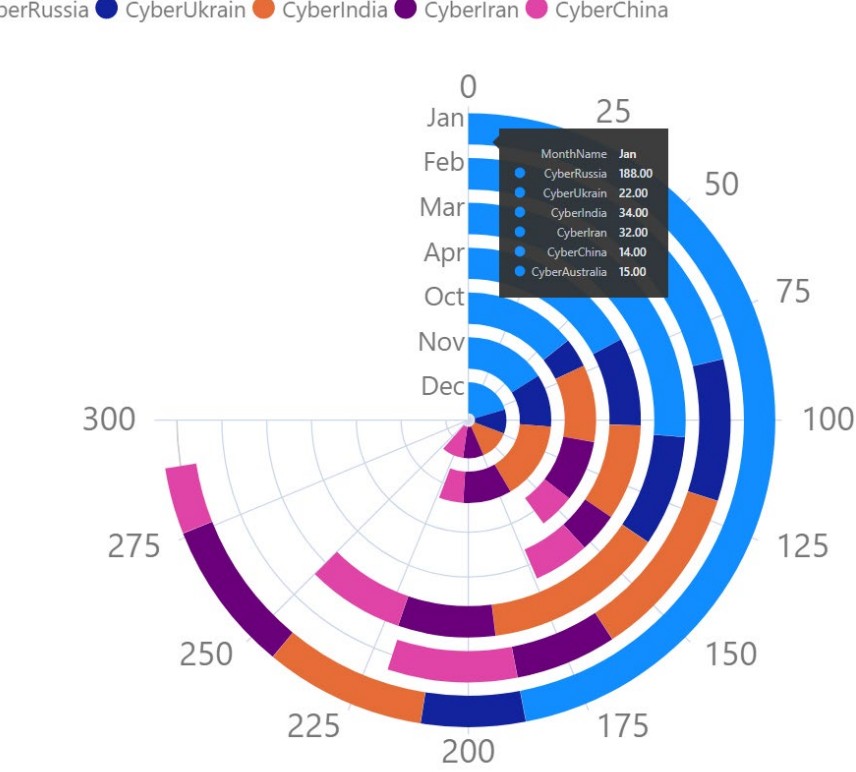

**Figure 9.** Ranking of number of cyber-related tweets for the selected countries by month (for the 7-month period).

**Table 8.** Performance of LDA algorithms for analysing topics on cyber-related tweets for China, Russia, Ukraine, India, and Australia.

| Performance Vectors | China | Russia | Ukraine | India | Australia |
|---|---|---|---|---|---|
| LogLikelihood | −15,617.27 | −57,933.967 | −23,251.897 | −27,119.332 | −9514.318 |
| Perplexity | 517.155 | 458.384 | 1016.203 | 759.998 | 322.952 |
| Avg(tokens) | 316.571 | 1165.143 | 392 | 519.857 | 206.143 |
| Avg(document_entropy) | 2.868 | 4.495 | 4.364 | 3.418 | 2.589 |
| Avg(word-length) | 5.857 | 6.143 | 7.229 | 5.8 | 7.286 |
| Avg(coherence) | −15.623 | −13.754 | −14.672 | −17.145 | −13.013 |
| Avg(uniform_dist) | 2.101 | 2.677 | 2.009 | 2.078 | 2.077 |
| Avg(corpus_dist) | 1.67 | 1.614 | 1.925 | 1.701 | 1.71 |
| Avg(eff_num_words) | 103.849 | 98.33 | 179.378 | 169.716 | 87.975 |
| Avg(token-doc-diff) | 0.005 | 0.001 | 0.007 | 0.003 | 0.008 |
| Avg(rank_1_docs) | 0.835 | 0.772 | 0.174 | 0.836 | 0.886 |
| Avg(allocation_count) | 0.876 | 0.85 | 0.16 | 0.864 | 0.901 |
| Avg(exclusivity) | 0.504 | 0.597 | 0.461 | 0.438 | 0.493 |
| AlphaSum | 0.091 | 0.118 | 8.434 | 0.1 | 0.058 |
| Beta | 0.285 | 0.127 | 0.642 | 0.272 | 0.26 |
| BetaSum | 378.828 | 386.22 | 1039.923 | 562.278 | 226.947 |

**Table 9.** Result of LDA algorithms for extracting 7 topics for China, Russia, Ukraine, India, and Australia.

| | TOPIC 1 | | TOPIC 2 | | TOPIC 3 | | TOPIC 4 | | TOPIC 5 | | TOPIC 6 | | TOPIC 7 | |
|---|---|---|---|---|---|---|---|---|---|---|---|---|---|---|
| **China** | cyber | 29 | China | 20 | China | 16 | Russia | 6 | China | 21 | China | 15 | China | 17 |
| | China | 22 | Cyber | 9 | Hack | 5 | China | 6 | hack | 12 | cyber | 7 | Chinese | 10 |
| | attacks | 14 | hack | 6 | country | 4 | North | 4 | chains | 4 | war | 6 | sophisticated | 8 |
| | Russia | 8 | TikTok | 4 | national | 3 | Cyber | 4 | supply | 4 | would | 6 | databases | 8 |
| | States | 7 | China's | 4 | IMMEDIATELY | 3 | reports | 3 | etc | 4 | Russia | 5 | Tech | 8 |
| **Russia** | Russian | 72 | cyber | 65 | Russian | 65 | hack | 70 | Russia | 60 | Russia | 97 | Russian | 60 |
| | cyber | 65 | Russian | 44 | hack | 25 | Russia | 36 | Invades | 50 | hacked | 19 | Putin | 59 |
| | attack | 49 | Ukraine | 24 | ShellenbergMD | 14 | Russian | 30 | Cyber | 42 | cyber | 18 | using | 58 |
| | blame | 27 | McGonigal | 22 | hacking | 14 | Russians | 27 | attacks | 33 | helped | 16 | Trump | 57 |
| | threat | 26 | FBI | 19 | amp | 13 | DNC | 16 | DarthPutinKGB | 26 | new | 16 | story | 57 |
| **Ukraine** | State | 3 | TheStudyofWar | 2 | says | 3 | role | 3 | country | 5 | Ukraine | 117 | Leaks | 2 |
| | absolutely | 2 | FBI | 2 | GicAriana | 2 | OMC_Ukraine | 2 | loser | 3 | cyber | 76 | cyberwarfare | 2 |
| | Threat | 2 | air | 2 | need | 2 | Anonymous_Link | 2 | brigade | 2 | Russian | 31 | cyberattacks | 2 |
| | report | 2 | infrastructure | 2 | don't | 2 | Council | 2 | hacker | 2 | Ukrainian | 28 | Red | 2 |
| | Cross | 2 | one | 2 | Security | 2 | UkraineRussiaWar | 2 | awareness | 2 | hack | 28 | never | 2 |
| **India** | hack | 9 | YouTube | 11 | Cyber | 55 | India | 19 | India | 29 | cyber | 17 | Cyber | 10 |
| | account | 9 | YouTubeIndia | 7 | Indian | 24 | cyber | 13 | cyber | 10 | India | 14 | India | 9 |
| | India | 8 | hack | 5 | cyber | 19 | Cyber | 11 | company | 8 | crime | 10 | amp | 8 |
| | IndiaFreeFire | 5 | Cyber | 5 | India | 18 | Indian | 9 | BJP | 6 | PMOIndia | 7 | Leaks | 3 |
| | please | 5 | YouTubeCreators | 4 | Crime | 13 | China | 7 | Hack | 5 | Cyber | 7 | BSF | 3 |
| **Australia** | Australians | 10 | Australian | 9 | Australia | 7 | cyber | 4 | cyber | 12 | amp | 7 | Police | 16 |
| | Australian | 9 | hack | 7 | way | 4 | POTUS | 3 | Australia | 11 | Australia | 7 | Australian | 14 |
| | scamming | 6 | Medibank | 6 | Cyber | 3 | Australia | 3 | data | 8 | Cyber | 5 | Cyber | 12 |
| | Boys | 6 | million | 5 | Australian | 3 | INSTAGRAM | 2 | Australian | 7 | Leaks | 3 | Australia | 10 |
| | Yahoo | 6 | health | 5 | fundamental | 2 | AustralianOpen | 2 | attack | 5 | https://t.co | 3 | love | 7 |

While Figures 6–9 represent the analysis of the social media agent and cognitive service agent from tweets, Figure 10 shows the actual deployment of the web media agent within a deployed mobile app (within Samsung Galaxy Note 10 Light mobile) analysing actual cyber-attack statistics from the web. As seen in Figure 10, the web media agent provided a ranking of the most cyber-attacked countries, and a user selected China (with an average rank factor of 67.74). Using this agent, it was identified that China was the most threatened country followed by Russia, Ukraine, India, Iran, and Australia (as shown in Figure 10). This information (of threat ranking) is used for the final assessment of "Threat Level" (second semantic output of the aggregation agent, as shown in Table 1).

Similarly, Figure 11 shows another deployment of a web media agent showing the cyber threat spectrum of the selected countries. As seen in Figure 11, the major cyber threat types for China, Russia, and India were spam. For Ukraine, Iran, and Australia, the threat types were deemed to be local infection, on-demand scans, and web threats, respectively. This information (i.e., shown in Figure 11) is subsequently used in explaining the "Threat Spectrum" by the aggregation agent (i.e., third semantic output of the aggregation agent,

as shown in Table 1). Finally, Figure 12 shows the web media agent analysing global cyber threats during the monitored period. Figure 12a shows the cyber threat prediction for Australia using exponential smoothing (described earlier in Section 3.6). Figure 12b demonstrates a detailed threat spectrum analysis for Russia. As seen in Figure 12b, in terms of spam, Russia appeared to be the third most attacked country in the world during the evaluation period. This information (i.e., shown in Figure 12b) is also used in explaining the "Threat Spectrum" by the aggregation agent (i.e., third semantic output of the aggregation agent, as shown in Table 1).

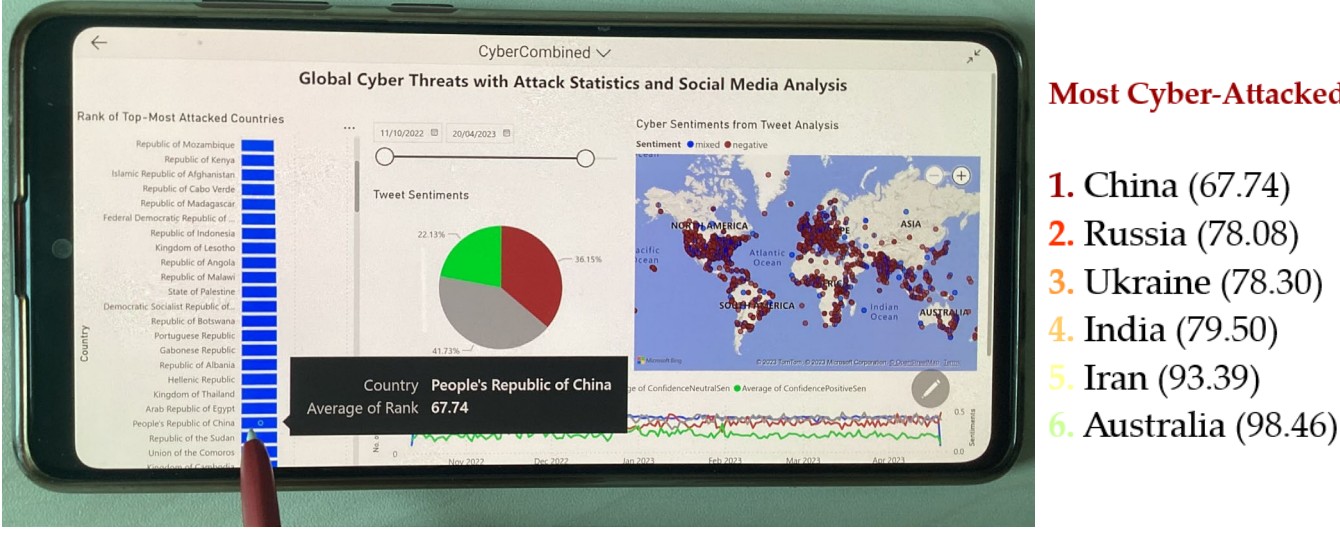

**Figure 10.** Ranking the most cyber-attacked countries (for the selected countries) using web cyber-attack statistics.

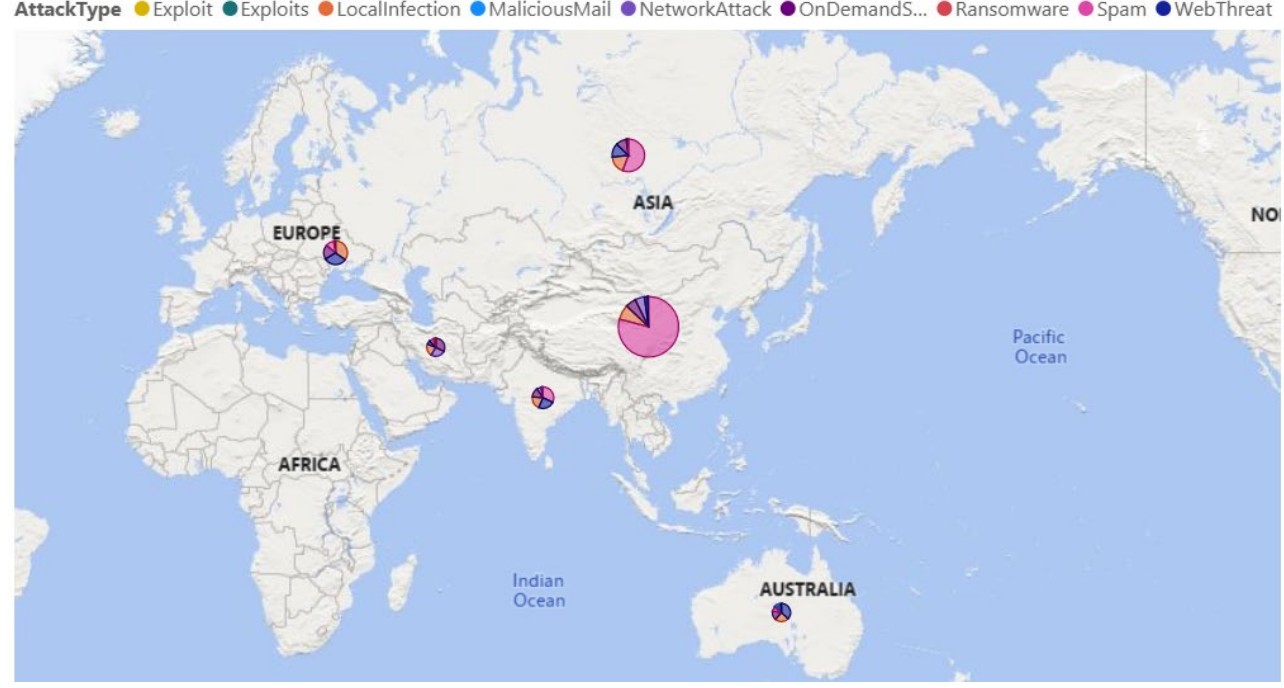

**Figure 11.** Identifying the threat types by web media agent using web-based cyber-attack statistics.

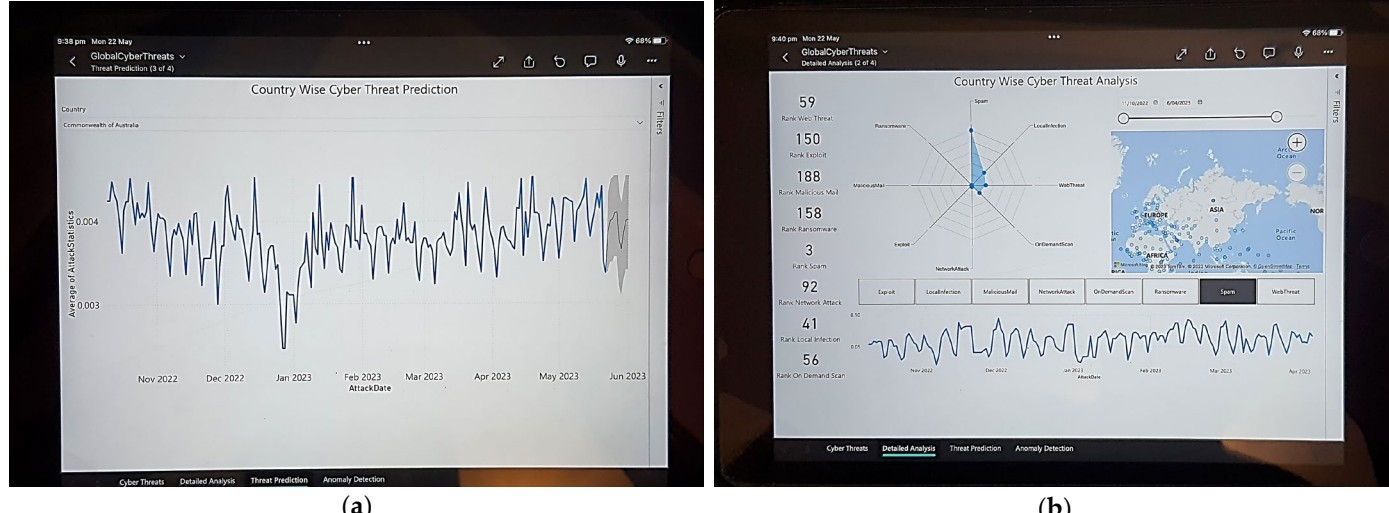

**Figure 12.** Web-media agent app deployed in iPad 9th Generation running iOS 15. (**a**) Cyber Threat Prediction for Australia. (**b**) Country wise detailed cyber threat analysis.

## 5. Discussion and Concluding Remarks

Even with multiple research mandates on developing innovative cyber threat agents, there have not been significant cyber agents reported in the existing literature [38]. This research developed a new agent-based cyber intelligence solution by deploying four independent agents. In this section, performance evaluation was performed in terms of how fast the solution is and how comprehensive the autonomous cyber report is.

The evaluation of the "fast" dimension is provided in Table 10. As seen in Table 10, the proposed solution with the aggregation agent, social media agent, cognitive service agent, and web media agent promoted the fastest response time.

**Table 10.** Evaluation of the agents with "fast" dimension.

| Number of Agents | Configuration of Agent | Average Response Time (Seconds) |
|---|---|---|
| One | A single agent processing both tweets and web-based cyber-attack statistics | 9.032 |
| Two | One agent processing tweets and another agent processing web-based cyber-attack statistics | 8.908 |
| Three | One agent performing aggregation, another one processing tweets, and the last agent processing web-based cyber-attack statistics | 7.781 |
| Four | One agent performing aggregation, two agents processing tweets, and the last agent processing web-based cyber-attack statistics (Proposed) | 6.451 |
| Five | One agent performing aggregation, two agents processing tweets, and the other two agents processing web-based cyber-attack statistics | 7.812 |

Table 11 provides an evaluation with respect to a comprehensive set of dimensions, such as threat level, threat spectrum, geopolitical/socioeconomic, psychological, impacted target, national concern, and victimization. Obtaining AI-driven information on these comprehensive lists of dimensions was the goal of the aggregation agent, as outlined in the semantic output (in Table 1). Interestingly, with the proposed configuration of agents, it is possible to obtain comprehensive cyber intelligence on any country within just 6 s (on average). It is imperative to emphasize that none of the extant solutions, namely, references [17–23], have achieved the level of comprehensive cyber intelligence by country as exemplified in this study. Table 11 serves as a demonstration of the proposed agent-based cyber intelligence solution's capacity and capability to generate thorough cyber intelligence pertaining to various countries, such as China, India, Russia, Ukraine, Australia, and Iran. The selection of these countries was random in nature, as the proposed system possesses the

inherent ability to generate cyber intelligence for any country across the globe, contingent upon the availability of social media users discussing said country.

**Table 11.** Evaluation of the agents with "comprehensive" dimension.

| Country Name | Threat Level | Threat Spectrum | Geopolitical | Psychological | Impacted Target | National Concern | Victimization |
|---|---|---|---|---|---|---|---|
| China | Deep Red | Spam, Network Attack | US, Russia | Moderate | TikTok, Database | Espionage, National Security | Supply Chain Tech Firms |
| Russia | Red | Spam | US, Russia | High | Putin, KGB | FBI, Trump | Putin, KGB, Russian Government |
| Ukraine | Deep Amber | Local Infection | Russia | High | Ukrainian Security | Ukraine Russia War | Infrastructure |
| India | Amber | Spam | China | Low | YouTube, BJP | YouTube Hack, Account Hack | Individual Accounts |
| Iran | Yellow | On-Demand Scan | US | Moderate | | | |
| Australia | Green | Web Threat | China | Moderate | Health (Medibank) Electricity Network | Data Breach, Malware, Phishing, Ransomware | Australian, Infrastructure |

In spite of proposing a novel and autonomous AI-driven semantic cyber agent, this study encounters several limitations and potential drawbacks.

- Firstly, the proposed approach assumed that all 37,386 cyber-related tweets were relevant. However, it is evident from the data presented in Table 12 that not all 37,386 tweets could be classified as cyber-related. Employing the confusion matrix depicted in Table 12, an array of performance evaluation criteria encompassing precision, recall, sensitivity, specificity, F1-score, accuracy, and others were computed and documented in Table 13. Upon comparing the performance of the proposed approach with existing research in the realm of social media-based cyber intelligence, it becomes apparent, as indicated in Table 14, that a few extant studies, specifically [17] and [21], outperformed the proposed approach in certain instances. Nonetheless, it is worth noting that the proposed approach exhibits superior performance compared to the majority of existing solutions documented in the literature. On average, the F1-score achieved by the prevailing methodologies was observed to be 0.83, whereas the proposed solution showcased a significantly higher F1-score of 0.88.
- Secondly, the proposed approach disregarded the possibility that these tweets could have originated from counterfeit accounts [39] or that genuine Twitter users may disseminate false information [40].
- Thirdly, this study relies on real-time tweet API, Microsoft Power Platform, and Microsoft Azure, all of which necessitate regular payment through credit cards. For instance, access to the basic Twitter API with a monthly limit of reading only 10K Tweets incurs a cost of $100 USD per month [41]. Increasing this limit to read 1 million tweets could result in a financial commitment of $5000 USD per month [41]. Consequently, in order to minimize expenses, this research examined only a limited number of tweets. Researchers interested in working with real-time tweets must possess access to credit cards and sufficient research funds to sustain the ongoing subscription costs.
- Fourthly, this research extensively employed "black box" cloud-based services and tools, such as Microsoft Cognitive Services, which poses substantial challenges in investigating algorithmic biases and potential enhancements.
- Lastly, this investigation employed industry standard tools and cutting-edge cloud services, including Microsoft Power Platform and Microsoft Azure. Therefore, conducting this research necessitates expertise and certifications in these technologies and standards.

**Table 12.** Confusion matrix in identifying relevancy of the cyber-related tweets.

|  | **Actual Positive** | **Actual Negative** |
|---|---|---|
| Predicted Positive | **23,178** (TP) | 2241 (FP) |
| Predicted Negative | 4149 (FN) | **7818** (TN) |

**Table 13.** Performance evaluations in identifying relevant cyber-related tweets.

| **Evaluation Metric** | **Formula** | **Calculation** |
|---|---|---|
| Precision | PPV = TP/(TP + FP) | 0.9118 |
| Recall | TPR = TP/(TP + FN) | 0.8482 |
| Sensitivity | TPR = TP/(TP + FN) | 0.8482 |
| Specificity | SPC = TN/(FP + TN) | 0.7772 |
| Negative Predictive Value | NPV = TN/(TN + FN) | 0.6533 |
| False Positive Rate | FPR = FP/(FP + TN) | 0.2228 |
| False Discovery Rate | FDR = FP/(FP + TP) | 0.0882 |
| False Negative Rate | FNR = FN/(FN + TP) | 0.1518 |
| Accuracy | ACC = (TP + TN)/(TP + FP + TN + FN) | 0.8291 |
| F1-Score | F1 = 2TP/(2TP + FP + FN) | 0.8789 |

**Table 14.** Performance comparison with existing research in social media-based cyber intelligence.

| **Algorithms Used** | **Precision** | **Recall** | **F1-Score** | **Reference** |
|---|---|---|---|---|
| Naïve Bayes (Negative) | 0.77 | 0.80 | 0.79 | [17] |
| Naïve Bayes (Positive) | 0.76 | 0.76 | 0.76 | [17] |
| Naïve Bayes (Security-Oriented) | 0.94 | 0.91 | 0.93 | [17] |
| Support Vector Machine (Negative) | 0.80 | 0.80 | 0.80 | [17] |
| Support Vector Machine (Positive) | 0.78 | 0.80 | 0.79 | [17] |
| Support Vector Machine (Security-Oriented) | 0.95 | 0.94 | 0.95 | [17] |
| Maximum Entropy (Negative) | 0.81 | 0.80 | 0.80 | [17] |
| Maximum Entropy (Positive) | 0.78 | 0.80 | 0.79 | [17] |
| Maximum Entropy (Security-Oriented) | 0.96 | 0.94 | 0.95 | [17] |
| Random Forest (CySecPriv) | 0.94 | 0.61 | 0.74 | [19] |
| Random Forest ('NonExpertUser) | 0.70 | 1.0 | 0.83 | [19] |
| LDA—VEM + TF-IDF (Personal) | - | - | 0.76 | [20] |
| LDA—VEM + TF-IDF (Professional) | - | - | 0.67 | [20] |
| LDA—VEM + TF-IDF (Health) | - | - | 0.75 | [20] |
| SVC (Cyber Bullying) | 0.73 | 0.96 | 0.83 | [21] |
| Logistic Regression (Cyber Bullying) | 0.91 | 0.96 | 0.93 | [21] |
| Multinomial Naïve Bayes (Cyber Bullying) | 0.86 | 0.94 | 0.90 | [21] |
| Random Forest Classifier (Cyber Bullying) | 0.98 | 0.73 | 0.84 | [21] |
| SGD Classifier (Cyber Bullying) | 0.90 | 0.95 | 0.93 | [21] |
| Light Gradient Boosted Machine (Darknet Traffic) | - | - | 0.84 | [22] |
| Proposed (Comprehensive Cyber) | 0.91 | 0.85 | 0.88 | |

In the future, our research endeavours will focus on the integration of advanced algorithms designed to identify and discern fake users, as well as counterfeit information, thereby facilitating the acquisition of more robust and validated cyber intelligence. Furthermore, we shall dedicate our efforts to further exploration of open-source tools and algorithms, aiming to mitigate cost obligations, enhance performance metrics (specifically in terms of F1-score), and foster a greater sense of technological autonomy.

**Funding:** This research received no external funding.

**Data Availability Statement:** Data will be made available upon request.

**Acknowledgments:** The author would like to acknowledge the feedback provided by Edris Alam of Disaster Resilience and Emergency Management, Rabdan Academy, Abu Dhabi, UAE.

**Conflicts of Interest:** The authors declare no conflict of interest.

## Appendix A

This section provides the Python code for implementing language detection and translation, sentiment analysis, and anomaly detection using the Microsoft Cognitive Services API.

*Appendix A.1. Language Detection and Translation*

Appendix A.1.1. Python Code Sample

```
import requests, uuid, json
# Add your key and endpoint
key = "<YOUR-TRANSLATOR-KEY>"
endpoint = "https://api.cognitive.microsofttranslator.com"
# location, also known as region.
# required if you're using a multi-service or regional (not global) resource. It can be found in the Azure portal on the Keys and Endpoint page.
location = "<YOUR-RESOURCE-LOCATION>"
path = '/translate'
constructed_url = endpoint + path
params = {
    'api-version': '3.0',
    'to': ['en', 'it']
}
headers = {
    'Ocp-Apim-Subscription-Key': key,
    # location required if you're using a multi-service or regional (not global) resource.
    'Ocp-Apim-Subscription-Region': location,
    'Content-type': 'application/json',
    'X-ClientTraceId': str(uuid.uuid4())
}
# You can pass more than one object in body.
body = [{
    'text': 'Halo, rafiki! Ulifanya nini leo?'
}]
request = requests.post(constructed_url, params=params, headers=headers, json=body)
response = request.json()
print(json.dumps(response, sort_keys=True, ensure_ascii=False, indent=4, separators=(',', ': ')))
```

Appendix A.1.2. Sample Output

```
[
    {
        "detectedLanguage":{
            "language":"sw",
            "score":0.8
        },
        "translations":[
            {
                "text":"Hello friend! What did you do today?",
                "to":"en"
            },
            {
                "text":"Ciao amico! Cosa hai fatto oggi?",
                "to":"it"
            }
```

```
            ]
        }
    ]
```

*Appendix A.2. Sentiment Analysis*

Appendix A.2.1. Python Code Sample

```
# This example requires environment variables named "LANGUAGE_KEY" and
"LANGUAGE_ENDPOINT"
language_key = os.environ.get('LANGUAGE_KEY')
language_endpoint = os.environ.get('LANGUAGE_ENDPOINT')

from azure.ai.textanalytics import TextAnalyticsClient
from azure.core.credentials import AzureKeyCredential

# Authenticate the client using your key and endpoint
def authenticate_client():
    ta_credential = AzureKeyCredential(language_key)
    text_analytics_client = TextAnalyticsClient(
            endpoint=language_endpoint,
            credential=ta_credential)
    return text_analytics_client
client = authenticate_client()
# Example method for detecting sentiment and opinions in text
def sentiment_analysis_with_opinion_mining_example(client):
    documents = [
        "The food and service were unacceptable. The concierge was nice, however".
    ]
    result = client.analyze_sentiment(documents, show_opinion_mining=True)
    doc_result = [doc for doc in result if not doc.is_error]
    positive_reviews = [doc for doc in doc_result if doc.sentiment == "positive"]
    negative_reviews = [doc for doc in doc_result if doc.sentiment == "negative"]
    positive_mined_opinions = []
    mixed_mined_opinions = []
    negative_mined_opinions = []
    for document in doc_result:
        print("Document Sentiment: {}".format(document.sentiment))
        print("Overall scores: positive={0:.2f}; neutral={1:.2f}; negative={2:.2f}
\n".format(
                document.confidence_scores.positive,
                document.confidence_scores.neutral,
                document.confidence_scores.negative,
        ))
        for sentence in document.sentences:
            print("Sentence: {}".format(sentence.text))
            print("Sentence sentiment: {}".format(sentence.sentiment))
            print("Sentence score:\nPositive={0:.2f}\nNeutral={1:.2f}\nNega-
tive={2:.2f}\n".format(
                    sentence.confidence_scores.positive,
                    sentence.confidence_scores.neutral,
                    sentence.confidence_scores.negative,
            ))
            for mined_opinion in sentence.mined_opinions:
                target = mined_opinion.target
                print("......'{}' target '{}'".format(target.sentiment, target.text))
```

```
                              print("......Target   score:\n......Positive={0:.2f}\n......Nega-
tive={1:.2f}\n".format(
                                  target.confidence_scores.positive,
                                  target.confidence_scores.negative,
                              ))
                          for assessment in mined_opinion.assessments:
                              print("......'{}' assessment '{}'".format(assessment.sentiment,
assessment.text))
                              print("......Assessment score:\n......Positive={0:.2f}\n......Nega-
tive={1:.2f} \n".format(
                                  assessment.confidence_scores.positive,
                                  assessment.confidence_scores.negative,
                              ))
                  print("\n")
              print("\n")

      sentiment_analysis_with_opinion_mining_example(client)
```

Appendix A.2.2. Sample Output

```
Document Sentiment: mixed
Overall scores: positive=0.47; neutral=0.00; negative=0.52
Sentence: The food and service were unacceptable.
Sentence sentiment: negative
Sentence score:
Positive=0.00
Neutral=0.00
Negative=0.99
......'negative' target 'food'
......Target score:
......Positive=0.00
......Negative=1.00
......'negative' assessment 'unacceptable'
......Assessment score:
......Positive=0.00
......Negative=1.00
......'negative' target 'service'
......Target score:
......Positive=0.00
......Negative=1.00
......'negative' assessment 'unacceptable'
......Assessment score:
......Positive=0.00
......Negative=1.00
Sentence: The concierge was nice, however.
Sentence sentiment: positive
Sentence score:
Positive=0.94
Neutral=0.01
Negative=0.05
......'positive' target 'concierge'
......Target score:
......Positive=1.00
......Negative=0.00
......'positive' assessment 'nice'
```

......Assessment score:
......Positive=1.00
......Negative=0.00

*Appendix A.3. Anomaly Detection*

Appendix A.3.1. Python Code

```
import time
from datetime import datetime, timezone
from azure.ai.anomalydetector import AnomalyDetectorClient
from azure.core.credentials import AzureKeyCredential
from azure.ai.anomalydetector.models import *
SUBSCRIPTION_KEY = os.environ['ANOMALY_DETECTOR_API_KEY']
ANOMALY_DETECTOR_ENDPOINT = os.environ['ANOMALY_DETEC-
TOR_ENDPOINT']
ad_client = AnomalyDetectorClient(ANOMALY_DETECTOR_ENDPOINT, AzureK-
eyCredential(SUBSCRIPTION_KEY))
time_format = "%Y-%m-%dT%H:%M:%SZ"
blob_url = "Path-to-sample-file-in-your-storage-account" # example path: https://
docstest001.blob.core.windows.net/test/sample_data_5_3000.csv
train_body = ModelInfo(
    data_source=blob_url,
    start_time=datetime.strptime("2021-01-02T00:00:00Z", time_format),
    end_time=datetime.strptime("2021-01-02T05:00:00Z", time_format),
    data_schema="OneTable",
    display_name="sample",
    sliding_window=200,
    align_policy=AlignPolicy(
        align_mode=AlignMode.OUTER,
        fill_n_a_method=FillNAMethod.LINEAR,
        padding_value=0,
    ),
)
batch_inference_body = MultivariateBatchDetectionOptions(
        data_source=blob_url,
        top_contributor_count=10,
        start_time=datetime.strptime("2021-01-02T00:00:00Z", time_format),
        end_time=datetime.strptime("2021-01-02T05:00:00Z", time_format),
    )
print("Training new model...(it may take a few minutes)")
model = ad_client.train_multivariate_model(train_body)
model_id = model.model_id
print("Training model id is {}".format(model_id))

## Wait until the model is ready. It usually takes several minutes
model_status = None
model = None
while model_status != ModelStatus.READY and model_status != ModelStatus.FAILED:
    model = ad_client.get_multivariate_model(model_id)
    print(model)
    model_status = model.model_info.status
    print("Model is {}".format(model_status))
    time.sleep(30)
if model_status == ModelStatus.READY:
    print("Done.\n--------------------")
```

```
        # Return the latest model id
        # Detect anomaly in the same data source (but a different interval)
        result = ad_client.detect_multivariate_batch_anomaly(model_id, batch_inference_body)
        result_id = result.result_id
        # Get results (may need a few seconds)
        r = ad_client.get_multivariate_batch_detection_result(result_id)
        print("Get detection result...(it may take a few seconds)")
        while r.summary.status != MultivariateBatchDetectionStatus.READY and r.summary.status
!= MultivariateBatchDetectionStatus.FAILED and r.summary.status !=MultivariateBatchDetec-
tionStatus.CREATED:
            anomaly_results = ad_client.get_multivariate_batch_detection_result(result_id)
            print("Detection is {}".format(r.summary.status))
            time.sleep(5)

        print("Result ID:\t", anomaly_results.result_id)
        print("Result status:\t", anomaly_results.summary.status)
        print("Result length:\t", len(anomaly_results.results))

        # See detailed inference result
        for r in anomaly_results.results:
            print(
                "timestamp: {}, is_anomaly: {:<5}, anomaly score: {:.4f}, severity: {:.4f},
contributor count: {:<4d}".format(
                    r.timestamp,
                    r.value.is_anomaly,
                    r.value.score,
                    r.value.severity,
                    len(r.value.interpretation) if r.value.is_anomaly else 0,
                )
            )
            if r.value.interpretation:
                for contributor in r.value.interpretation:
                    print(
                        "\tcontributor variable: {:<10}, contributor score: {:.4f}".format(
                            contributor.variable, contributor.contribution_score
                        )
                    )
```

## Appendix A.3.2. Sample Output

10 available models before training.
Training new model...(it may take a few minutes)
Training model id is 3a695878-a88f-11ed-a16c-b290e72010e0
{'modelId': '3a695878-a88f-11ed-a16c-b290e72010e0', 'createdTime': '2023-02-09T15:34:23Z', 'lastUpdatedTime': '2023-02-09T15:34:23Z', 'modelInfo': {'dataSource': 'https://docstest0 01.blob.core.windows.net/test/sample_data_5_3000 (1).csv', 'dataSchema': 'OneTable', 'startTime': '2021-01-02T00:00:00Z', 'endTime': '2021-01-02T05:00:00Z', 'displayName': 'sample', 'slidingWindow': 200, 'alignPolicy': {'alignMode': 'Outer', 'fillNAMethod': 'Linear', 'paddingValue': 0.0}, 'status': 'CREATED', 'errors': [], 'diagnosticsInfo': {'modelState': {'epochIds': [], 'trainLosses': [], 'validationLosses': [], 'latenciesInSeconds': []}, 'variableStates': []}}}
Model is CREATED
{'modelId': '3a695878-a88f-11ed-a16c-b290e72010e0', 'createdTime': '2023-02-09T15:34:23Z', 'lastUpdatedTime': '2023-02-09T15:34:55Z', 'modelInfo': {'dataSource': 'https://docstest0

01.blob.core.windows.net/test/sample_data_5_3000 (1).csv', 'dataSchema': 'OneTable', 'startTime': '2021-01-02T00:00:00Z', 'endTime': '2021-01-02T05:00:00Z', 'displayName': 'sample', 'slidingWindow': 200, 'alignPolicy': {'alignMode': 'Outer', 'fillNAMethod': 'Linear', 'paddingValue': 0.0}, 'status': 'READY', 'errors': [], 'diagnosticsInfo': {'modelState': {'epochIds': [10, 20, 30, 40, 50, 60, 70, 80, 90, 100], 'trainLosses': [1.0493712276220322, 0.5454281121492386, 0.42524269968271255, 0.38019897043704987, 0.3472398854792118, 0.34301353991031647, 0.3219067454338074, 0.3108387663960457, 0.30357857793569565, 0.29986055195331573], 'validationLosses': [0.0, 0.0, 0.0, 0.0, 0.0, 0.0, 0.0, 0.0, 0.0, 0.0], 'latenciesInSeconds': [0.3412797451019287, 0.25798678398132324, 0.2556419372558594, 0.3165152072906494, 0.2748451232910156, 0.26111531257629395, 0.2571413516998291, 0.257282018661499, 0.2549862861633301, 0.25806593894958496]}, 'variableStates': [{'variable': 'series_0', 'filledNARatio': 0.0, 'effectiveCount': 301, 'firstTimestamp': '2021-01-02T00:00:00Z', 'lastTimestamp': '2021-01-02T05:00:00Z'}, {'variable': 'series_1', 'filledNARatio': 0.0, 'effectiveCount': 301, 'firstTimestamp': '2021-01-02T00:00:00Z', 'lastTimestamp': '2021-01-02T05:00:00Z'}, {'variable': 'series_2', 'filledNARatio': 0.0, 'effectiveCount': 301, 'firstTimestamp': '2021-01-02T00:00:00Z', 'lastTimestamp': '2021-01-02T05:00:00Z'}, {'variable': 'series_3', 'filledNARatio': 0.0, 'effectiveCount': 301, 'firstTimestamp': '2021-01-02T00:00:00Z', 'lastTimestamp': '2021-01-02T05:00:00Z'}, {'variable': 'series_4', 'filledNARatio': 0.0, 'effectiveCount': 301, 'firstTimestamp': '2021-01-02T00:00:00Z', 'lastTimestamp': '2021-01-02T05:00:00Z'}]}}}}

Model is READY
Done.
--------------------
10 available models after training.
Get detection result...(it may take a few seconds)
Detection is CREATED
Detection is READY
Result ID: 70a6cdf8-a88f-11ed-a461-928899e62c38
Result status: READY
Result length: 301
timestamp: 2021-01-02 00:00:00+00:00, is_anomaly: 0 , anomaly score: 0.1770, severity: 0.0000, contributor count: 0
timestamp: 2021-01-02 00:01:00+00:00, is_anomaly: 0 , anomaly score: 0.3446, severity: 0.0000, contributor count: 0
timestamp: 2021-01-02 00:02:00+00:00, is_anomaly: 0 , anomaly score: 0.2397, severity: 0.0000, contributor count: 0
timestamp: 2021-01-02 00:03:00+00:00, is_anomaly: 0 , anomaly score: 0.1270, severity: 0.0000, contributor count: 0
timestamp: 2021-01-02 00:04:00+00:00, is_anomaly: 0 , anomaly score: 0.3321, severity: 0.0000, contributor count: 0
timestamp: 2021-01-02 00:05:00+00:00, is_anomaly: 0 , anomaly score: 0.4053, severity: 0.0000, contributor count: 0
timestamp: 2021-01-02 00:06:00+00:00, is_anomaly: 0 , anomaly score: 0.4371, severity: 0.0000, contributor count: 0
timestamp: 2021-01-02 00:07:00+00:00, is_anomaly: 1 , anomaly score: 0.6615, severity: 0.3850, contributor count: 5
contributor variable: series_3 , contributor score: 0.2939
contributor variable: series_1 , contributor score: 0.2834
contributor variable: series_4 , contributor score: 0.2329
contributor variable: series_0 , contributor score: 0.1543
contributor variable: series_2 , contributor score: 0.0354

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
