# Peer review of "A New AI-Based Semantic Cyber Intelligence Agent"

_futureinternet, doi:10.3390/fi15070231_

Round 1

Reviewer 1 Report

This paper presents an innovative concept in the sphere of Cyber-Intelligence, exploring the use of a multi-agent system for the generation of comprehensive intelligence from social media data and web-based data. The intelligence output derived from the agent is framed to be beneficial for cyber analysts and strategists, thereby positioning the paper's premise in an undoubtedly practical and timely light. The feature of generating a user-friendly cyber-intelligence report amalgamating insights from various AI-driven sources is commendable, offering users a more holistic view of the digital landscape.

The paper's main novelty lies in its approach of reusing mature tools and models, signifying an efficient and practical use of existing resources. However, this strength is also potentially its biggest weakness, as the paper fails to provide adequate details about these tools and models. Considering that these tools and models are crucial components of the proposed multi-agent system, it is imperative for readers and researchers to understand how they operate and interplay within the system. This lack of transparency might limit the applicability and reproducibility of the system.

Issues related to the use of Azure Cognitive Services and other "black box" tools are prominently highlighted in the paper. While these tools might provide a convenient and efficient means of data processing and analysis, their use inherently inhibits the ability to control the quality of inference, false positives, and bias. The lack of transparency and control over these tools might raise concerns about the validity and reliability of the generated cyber-intelligence.

The presentation of Figure 1 should be improved. 

The application of Named Entity Recognition (NER) to the collected data is mentioned, but there is no elaboration on how the named-entities are extracted, what tools are utilized, or any evaluation on the effectiveness of the NER tool. This leaves readers in the dark about a crucial part of the data processing pipeline.

The details about the dataset used in the study, consisting of 37,386 Tweets from 30,706 Users in 54 languages collected from October 13, 2022, to April 6, 2023, are insufficient. There is no explanation about the source of the dataset, the collection process, or the methodology for separating relevant tweets from irrelevant ones.

The same applies to the 238,220 cyber-threat data obtained and processed by the web-media agent during the same timeframe. The details of the web-crawling process, the strategy for distinguishing between relevant and irrelevant pages, and any other data cleaning steps are not articulated.

Regarding the deployment of the so called "agents", it is mentioned that they were deployed, but there are no details about the configuration, whether the deployment was on cloud or locally, or any other specifics of the deployment process.

In light of these issues, it is apparent that the paper requires a comprehensive list of limitations and future work. Addressing these issues and providing more detailed explanations would significantly enhance the quality, transparency, and reproducibility of the research, allowing it to make a stronger contribution to the field of Cyber-Intelligence.

Author Response

First of all, I would like to express the honorable reviewer for taking an interest in this paper. I am glad that the expert reviewer highlighted this approach as innovative referring the presented Agent-based Cyber-Intelligence solution as innovative, beneficial, practical, timely, and holistic.

I have to praise the following 9 comments / suggestion of the reviewer as all of them are valid and it represents the in-depth knowledge of the reviewer in this subject matter. Hence, I have addressed all of them in details. I have no doubt that addressing these valuable suggestions of the expert reviewer has significantly enhanced the overall quality of the updated manuscript.

Comment / Suggestion 1: This paper presents an innovative concept in the sphere of Cyber-Intelligence, exploring the use of a multi-agent system for the generation of comprehensive intelligence from social media data and web-based data. The intelligence output derived from the agent is framed to be beneficial for cyber analysts and strategists, thereby positioning the paper's premise in an undoubtedly practical and timely light. The feature of generating a user-friendly cyber-intelligence report amalgamating insights from various AI-driven sources is commendable, offering users a more holistic view of the digital landscape.

Many thanks to the expert reviewer’s encouraging comments. This comment perfectly summarizes the presented system. Indeed, the system provides a holistic view of Cyber-Intelligence with innovative multi-agents and it is both practical and timely.

Comment / Suggestion 2: The paper's main novelty lies in its approach of reusing mature tools and models, signifying an efficient and practical use of existing resources. However, this strength is also potentially its biggest weakness, as the paper fails to provide adequate details about these tools and models. Considering that these tools and models are crucial components of the proposed multi-agent system, it is imperative for readers and researchers to understand how they operate and interplay within the system. This lack of transparency might limit the applicability and reproducibility of the system.

Many thanks for this suggestion. I completely agree with the honorable reviewer’s view on this. Taking this valuable suggestion constructively, I have added several paragraphs of implementation details specifically on critical area of the proposed system such as “Language Detection & Translation” (section 3.1), “Sentiment Analysis” (section 3.2), Anomaly Detection (section 3.3) and others. These paragraphs provide step by step process of implementing NLP models such as translation and sentiment analysis. Moreover, I have added Appendix section (from Line 477 to 800) showing sample python codes and corresponding outputs for enhancing transparency and reproducibility of the system.

Comment / Suggestion 3: Issues related to the use of Azure Cognitive Services and other "black box" tools are prominently highlighted in the paper. While these tools might provide a convenient and efficient means of data processing and analysis, their use inherently inhibits the ability to control the quality of inference, false positives, and bias. The lack of transparency and control over these tools might raise concerns about the validity and reliability of the generated cyber-intelligence.

I concur with the valid point raised by the expert reviewer. In the updated manuscript, I have now added three new Tables (i.e., Table 12, Table 13, and Table 14) to evaluate the effectiveness of the proposed solution in terms of false positives, false negatives, recall, precision, F1-Score and other metrics. Moreover, I have now compared the performance of the proposed solution with existing methods (that appeared in Table 3). These newly added tables (i.e., Table 12, Table 13, and Table 14) will provide a benchmark and help in assessing the superiority or effectiveness of the proposed approach.

While “black box” tools are known to provide limited control over quality, false positive, and transparency, research community have demonstrated innovative use of “black box” tools in solving complex research problems. There are many papers in top scientific journals, where the use of “black box” algorithms have successfully been reported. Moreover, researchers evaluated the results of these “black box” tool or algorithms specifically focusing on true positives, true negatives, false positives, and false negatives and calculating detailed precision, recall, F1-Score, and accuracies. For example, following of my recent publications all used black box algorithms like Azure Cognitive Service [1-13]. My research in [1] used Azure Cognitive Service based Sentiment Analysis, Named Entity Recognition on global events and obtained 0.992 precision, 0.993 recall, 0.993 F1-score, and 0.985 accuracy. Similarly, my research reported in [5] (Q1, Impact Factor 5.23) used Azure Cognitive Services’ Sentiment Analysis, Named Entity Recognition, and Category Classification on natural disaster related social media posts. The rates of average precision, recall, and F1-Score were measured to be 0.93, 0.88, and 0.90, respectively in [5] demonstrating higher performance of these black box algorithm. Similarly, in [11] black box algorithms like Azure Cognitive Service was used on political tweets to obtain political threat. The overall performance of the proposed system was noted as 0.90 precision rate, 0.90 recall rate, 0.90 F1 score, and 0.97 accuracy in [11].

[1] Fahim K. Sufi, Musleh Alsulami and Adnan Gutub, Automating Global Threat-Maps Generation via Advancements of News Sensors and AI, Arabian Journal for Science and Engineering (Springer), Vol. 48, PP. 2455–2472, 2023 (IF: 2.807)

[2] Fahim Sufi, Algorithms in Low-Code-No-Code for Research Applications: A Practical Review, Algorithms, Vol. 16, No. 2. 108, DOI: https://doi.org/10.3390/a16020108, 2023

[3] Fahim Sufi, Automatic identification and explanation of root causes on COVID-19 index anomalies, MethodsX (Elsevier), Vol. 10, No. 101960, DOI: https://doi.org/10.1016/j.mex.2022.101960, 2023

[4] Fahim Sufi, A New Social Media-Driven Cyber Threat Intelligence, Electronics, Vol. 12. No. 5, PP. 1242, https://doi.org/10.3390/electronics12051242, 2023 (IF: 2.690)

[5] Fahim Sufi and Ibrahim Khalil, Automated Disaster Monitoring from Social Media Posts using AI based Location Intelligence and Sentiment Analysis, IEEE Transactions on Computational Social Systems, (Accepted, in Press DOI: https://doi.org/10.1109/TCSS.2022.3157142), 2022 (IF: 5.23, Q1)

[6] Fahim Sufi, Imran Razzak and Ibrahim Khalil, Tracking Anti-Vax Social Movement Using AI based Social Media Monitoring, IEEE Transactions on Technology and Society, Vol. 3, No. 4, PP. pp. 290-299, https://doi.org/10.1109/TTS.2022.3192757, 2022

[7] Fahim Sufi, A decision support system for extracting artificial intelligence-driven insights from live twitter feeds on natural disasters, Decision Analytics Journal (Elsevier), Vol. 5, No. 100130, DOI: https://doi.org/10.1016/j.dajour.2022.100130, 2022

[8] Fahim Sufi, “AI-SocialDisaster: An AI-based software for identifying and analyzing natural disasters from social media”, Software Impacts (Elsevier), Vol 11, No 100319, 2022, DOI: https://doi.org/10.1016/j.simpa.2022.100319

[9] F. Sufi and M. Alsulami, "AI-based Automated Extraction of Location-Oriented COVID-19 Sentiments," Computers, Materials & Continua (CMC), Vols. 72, no. 2, pp. 3631–3649, 2022. DOI: https://doi.org/10.32604/cmc.2022.026272  (IF: 3.772, Q1)

[10] Fahim Sufi, Identifying the Drivers of Negative News with Sentiment, Entity and Regression Analysis, International Journal of Information Management Data Insights (Elsevier), Vol. 2, No. 1, 100074, 2022, DOI: https://doi.org/10.1016/j.jjimei.2022.100074

[11] F. Sufi and M. Alsulami, "A Novel Method of Generating Geospatial Intelligence from Social Media Posts of Political Leaders," Information, vol. 13, no. 3, p. 120, https://doi.org/10.3390/info13030120,  2022.

[12] Fahim Sufi, AI-GlobalEvents: A Software for analyzing, identifying and explaining global events with Artificial Intelligence, Software Impacts (Elsevier), Vol 11, No 100218, 2022, DOI: https://doi.org/10.1016/j.simpa.2022.100218

[13] Fahim Sufi and M. Alsulami, "Automated Multidimensional Analysis of Global Events with Entity Detection, Sentiment Analysis and Anomaly Detection," IEEE Access, Vol. 9, 2021, DOI: https://ieeexplore.ieee.org/document/9612169 (IF: 3.367, Q1)

Comment / Suggestion 4: The presentation of Figure 1 should be improved. 

Many thanks for raising this issue. I agree, Figure 1 looked a bit blurry. Hence, in the updated manuscript, I have improved the quality of Figure 1 and it is now in 330 PPI (i.e., HD Quality).

Comment / Suggestion 5: The application of Named Entity Recognition (NER) to the collected data is mentioned, but there is no elaboration on how the named-entities are extracted, what tools are utilized, or any evaluation on the effectiveness of the NER tool. This leaves readers in the dark about a crucial part of the data processing pipeline.

I used NER using Microsoft Cognitive Service in several of my recent research on Natural Disaster Analysis, Global Event Analysis, COVID-19 Analysis, and Political Threat Analysis from social media messages as shown below:

[1] Fahim K. Sufi, Musleh Alsulami and Adnan Gutub, Automating Global Threat-Maps Generation via Advancements of News Sensors and AI, Arabian Journal for Science and Engineering (Springer), Vol. 48, PP. 2455–2472, 2023 (IF: 2.807)

[2] Fahim Sufi, Automatic identification and explanation of root causes on COVID-19 index anomalies, MethodsX (Elsevier), Vol. 10, No. 101960, DOI: https://doi.org/10.1016/j.mex.2022.101960, 2023

[3] Fahim Sufi and Ibrahim Khalil, Automated Disaster Monitoring from Social Media Posts using AI based Location Intelligence and Sentiment Analysis, IEEE Transactions on Computational Social Systems, (Accepted, in Press DOI: https://doi.org/10.1109/TCSS.2022.3157142), 2022 (IF: 5.23, Q1)

[4] Fahim Sufi, A decision support system for extracting artificial intelligence-driven insights from live twitter feeds on natural disasters, Decision Analytics Journal (Elsevier), Vol. 5, No. 100130, DOI: https://doi.org/10.1016/j.dajour.2022.100130, 2022

[5] Fahim Sufi, “AI-SocialDisaster: An AI-based software for identifying and analyzing natural disasters from social media”, Software Impacts (Elsevier), Vol 11, No 100319, 2022, DOI: https://doi.org/10.1016/j.simpa.2022.100319

[6] F. Sufi and M. Alsulami, "AI-based Automated Extraction of Location-Oriented COVID-19 Sentiments," Computers, Materials & Continua (CMC), Vols. 72, no. 2, pp. 3631–3649, 2022. DOI: https://doi.org/10.32604/cmc.2022.026272  (IF: 3.772, Q1)

[7] Fahim Sufi, Identifying the Drivers of Negative News with Sentiment, Entity and Regression Analysis, International Journal of Information Management Data Insights (Elsevier), Vol. 2, No. 1, 100074, 2022, DOI: https://doi.org/10.1016/j.jjimei.2022.100074

[8] F. Sufi and M. Alsulami, "A Novel Method of Generating Geospatial Intelligence from Social Media Posts of Political Leaders," Information, vol. 13, no. 3, p. 120, https://doi.org/10.3390/info13030120,  2022.

[9] Fahim Sufi, AI-GlobalEvents: A Software for analyzing, identifying and explaining global events with Artificial Intelligence, Software Impacts (Elsevier), Vol 11, No 100218, 2022, DOI: https://doi.org/10.1016/j.simpa.2022.100218

[10] Fahim Sufi and M. Alsulami, "Automated Multidimensional Analysis of Global Events with Entity Detection, Sentiment Analysis and Anomaly Detection," IEEE Access, Vol. 9, 2021, DOI: https://ieeexplore.ieee.org/document/9612169 (IF: 3.367, Q1)

However, this paper does not use NER. In the updated manuscript, I have clarified the implementational details of using Microsoft Cognitive Services for implementing Sentiment Analysis, Language Detection, and Language Translation. Moreover, the appendix section now provides sample codes to invoke Microsoft Cognitive Services API’s. If a researcher wants to invoke NER from Microsoft Cognitive Services, then the process is similar.

Comment / Suggestion 6: The details about the dataset used in the study, consisting of 37,386 Tweets from 30,706 Users in 54 languages collected from October 13, 2022, to April 6, 2023, are insufficient. There is no explanation about the source of the dataset, the collection process, or the methodology for separating relevant tweets from irrelevant ones.

I agree with this valuable observation. Therefore, I have added the following new paragraph explaining the source of the Tweets as well as collection process (i.e., methodology).

“These Tweets were obtained using Microsoft Power Automate as demonstrated in Fig. 5. Acquiring tweets through Power Automate involves leveraging its capabilities to integrate with external platforms, such as Twitter's Application Programming Interface (API) as demonstrated in our research [16] [33] [34]. Power Automate, a cloud-based service pro-vided by Microsoft, allows users to automate workflows and create custom applications without requiring extensive coding knowledge. To obtain tweets, one must first authenticate their Power Automate connection with the Twitter API by generating API credentials and obtaining an access token. This process involves registering a Twitter developer ac-count, creating an application, and obtaining the necessary keys and tokens. Once authenticated, users can utilize Power Automate's "HTTP" action to send requests to the Twitter API's endpoints, such as the "search/tweets" endpoint. By specifying relevant parameters, such as search keywords, date range, or user handles, users can retrieve specific tweets or perform comprehensive searches. Power Automate can then process and manipulate the received tweet data, enabling users to store them in a database, send notifications, perform sentiment analysis, or integrate them into other applications, thereby empowering users to streamline their workflows and leverage Twitter's vast data resources. It should be mentioned that keywords like as “Cyber” and “Hack” were used to obtain the real-time Tweets from October 13, 2022, to April 6, 2023. All Tweets (regardless of their relevancy) were analyzed the proposed system.”

Comment / Suggestion 7: The same applies to the 238,220 cyber-threat data obtained and processed by the web-media agent during the same timeframe. The details of the web-crawling process, the strategy for distinguishing between relevant and irrelevant pages, and any other data cleaning steps are not articulated.

Many thanks for raising this concern. Since web media contains a plethora of information on cyber-threat statistics that might not be relevant, we obtained data only from validated and trusted sources such as Kaspersky. There were 8 specific URLs, that were used by the web-crawling process to obtain daily statistics of 8 dimensions of cyber threats (e.g., ransomware, vulnerability, web-threat, spam, malicious mail, network-attack etc.). Data accumulated on a daily basis on these 8 URLs from October 13, 2022, to April 6, 2023, resulted in 238,220 cyber-threat data. 

I have now clarified this with the following newly added paragraph within section 4:

“Kaspersky being a trusted provider of cyber-threat statistics, following multi-dimensional cyber-attack data were obtained by Web-Media agent with web scraping techniques:

  • Daily Ransomware data from https://statistics.securelist.com/ransomware/day
  • Daily Vulnerability data from https://statistics.securelist.com/vulnerability-scan/day
  • Daily Web-Threat data from https://statistics.securelist.com/web-anti-virus/day
  • Daily Spam data from https://statistics.securelist.com/kaspersky-anti-spam/day
  • Daily Malicious Mail data from https://statistics.securelist.com/mail-anti-virus/day
  • Daily Network-Attack data from https://statistics.securelist.com/intrusion-detection-scan/day
  • Daily Local Infection data from https://statistics.securelist.com/on-access-scan/day
  • Daily On-demand-scan data from https://statistics.securelist.com/on-demand-scan/day

In should be highlighted that all these multi-dimensional cyber-threat statistics from Kaspersky’s attack statistics (i.e., https://statistics.securelist.com/) site provided daily dump of threat statistics. Cloud-based Web-Media agent built with Microsoft Power Automate downloaded these statistics at daily schedule and saved these statistics within Microsoft Dataverse as demonstrated in Figure 5. The low-code implementation of technique is also demonstrated in [18].”

Comment / Suggestion 8: Regarding the deployment of the so called "agents", it is mentioned that they were deployed, but there are no details about the configuration, whether the deployment was on cloud or locally, or any other specifics of the deployment process.

Many thanks for raising this valid point. I concur with this critical observation and accordingly, I have added the following paragraph detailing agent configuration.

“Could-based Microsoft Power Platform and Microsoft Azure ecosystem were used for deploying the agents. Hence, industry standard configurations were adopted for agent deployment. Agents in Microsoft Azure are deployed through Azure Automation, a cloud-based service facilitating the automation and orchestration of tasks across Azure resources and external systems. To initiate agent deployment, an Azure Automation account is created as the central management hub. Subsequently, the Azure Automation agent is installed on the target machine or virtual machine, enabling the execution of automation runbooks and establishing secure communication between the agent and the automation service. Configuration of agent settings, such as defining runbook worker groups, proxy settings, network access control rules, and resource management, follows the agent's successful connection to the Azure Automation account. By assigning runbooks to the deployed agent within the Azure Automation account, desired automation tasks or workflows can be executed on the target machine. This deployment process empowers users to streamline their Azure resources through effective task automation and orchestration.”

Comment / Suggestion 9: In light of these issues, it is apparent that the paper requires a comprehensive list of limitations and future work. Addressing these issues and providing more detailed explanations would significantly enhance the quality, transparency, and reproducibility of the research, allowing it to make a stronger contribution to the field of Cyber-Intelligence.

I completely agree with this suggestion. Therefore, I have now added the following paragraph,

“In spite of proposing a novel and autonomous AI-driven semantic cyber agent, this study encounters several limitations and potential drawbacks.

  • Firstly, the proposed approach assumed that all 37,386 cyber-related Tweets were relevant. However, it is evident from the data presented in Table 12 that not all 37,386 Tweets could be classified as cyber-related. Employing the confusion matrix depicted in Table 12, an array of performance evaluation criteria, encompassing precision, re-call, sensitivity, specificity, F1-Score, accuracy, and others, were computed and documented in Table 13. Upon comparing the performance of the proposed approach with existing research in the realm of social media-based cyber intelligence, it be-comes apparent, as indicated in Table 14, that a few extant studies, specifically [17] and [21], outperformed the proposed approach in certain instances. Nonetheless, it is worth noting that the proposed approach exhibits superior performance compared to the majority of existing solutions documented in the literature. On average, the F1-Score achieved by the prevailing methodologies was observed to be 0.83, whereas the proposed solution showcased a significantly higher F1-Score of 0.88.
  • Secondly, the proposed approach disregarded the possibility that these Tweets could have originated from counterfeit accounts [39], or that genuine Twitter users may disseminate false information [40].
  • Thirdly, the study relies on real-time Tweet API, Microsoft Power Platform, and Microsoft Azure, all of which necessitate regular payment through credit cards. For in-stance, access to the basic Tweeter API with a monthly limit of reading only 10K Tweets incurs a cost of $100 USD per month [41]. Increasing this limit to read 1 mil-lion Tweets could result in a financial commitment of $5000 USD per month [41]. Consequently, in order to minimize expenses, this research examined only a limited number of Tweets. Researchers interested in working with real-time Tweets must possess access to credit cards and sufficient research funds to sustain the ongoing subscription costs.
  • Fourthly, this research extensively employed "black box" cloud-based services and tools, such as Microsoft Cognitive Services, which poses substantial challenges in investigating algorithmic biases and potential enhancements.
  • Lastly, this investigation employed industry-standard tools and cutting-edge cloud services, including Microsoft Power Platform and Microsoft Azure. Therefore, conducting this research necessitates expertise and certifications on these technologies and standards.

In future, our research endeavors will focus on the integration of advanced algorithms designed to identify and discern fake users, as well as counterfeit information, thereby facilitating the acquisition of more robust and validated cyber intelligence. Furthermore, we shall dedicate our efforts to further exploration of open-source tools and algorithms, aiming to mitigate cost obligations, enhance performance metrics (specifically in terms of F1-Score), and foster a greater sense of technological autonomy.”

Reviewer 2 Report

The paper proposes a novel cyber intelligence solution, which employs four semantic agents that interact autonomously to acquire crucial cyber intelligence pertaining to any given country. The solution leverages a combination of techniques, including Convolutional Neural Network (CNN), sentiment analysis, exponential smoothing, Latent Dirichlet Allocation (LDA), Term Frequency-Inverse Document Frequency (TF-IDF), Porter Stemming, and others, to analyze data from both social media and web sources.

While the paper has a good structure, there are several issues that need to be addressed:

1) Lack of discussion on the disadvantages of the proposed method: It is important to provide a balanced view by discussing the limitations or potential drawbacks of the proposed solution. This will enhance the credibility of the research and provide insights for future improvements.

2) Table 3 lacks a definition of notation (x): It is crucial to provide clear definitions of any notations used in the table for readers' understanding. This will help in comprehending the presented data accurately.

3) Lack of comparison of the proposed method's results with other reference methods from Table 3: To evaluate the effectiveness of the proposed solution, it is essential to compare its performance with existing reference methods. This comparison will provide a benchmark and help in assessing the superiority or effectiveness of the proposed approach.

Should fix small typos

Author Response

Comment / Suggestion 1: The paper proposes a novel cyber intelligence solution, which employs four semantic agents that interact autonomously to acquire crucial cyber intelligence pertaining to any given country. The solution leverages a combination of techniques, including Convolutional Neural Network (CNN), sentiment analysis, exponential smoothing, Latent Dirichlet Allocation (LDA), Term Frequency-Inverse Document Frequency (TF-IDF), Porter Stemming, and others, to analyze data from both social media and web sources.

While the paper has a good structure, there are several issues that need to be addressed:

First of all, I would like to thank the honorable reviewer for appreciating the novelty the proposed approach. I am also pleased to know that the honorable reviewer liked the overall structure of the paper. I have to praise comments / suggestion of the reviewer as all of them are valid and it represents the in-depth knowledge of the reviewer in this subject matter. I found all the suggestions valuable and addressed them in the updated manuscript. I have no doubt that addressing these valuable suggestions of the expert reviewer has significantly enhanced the overall quality of the updated manuscript.   

Comment / Suggestion 2: Lack of discussion on the disadvantages of the proposed method: It is important to provide a balanced view by discussing the limitations or potential drawbacks of the proposed solution. This will enhance the credibility of the research and provide insights for future improvements.

I completely agree with this suggestion. Therefore, I have now added the following paragraph,

“In spite of proposing a novel and autonomous AI-driven semantic cyber agent, this study encounters several limitations and potential drawbacks.

  • Firstly, the proposed approach assumed that all 37,386 cyber-related Tweets were relevant. However, it is evident from the data presented in Table 12 that not all 37,386 Tweets could be classified as cyber-related. Employing the confusion matrix depicted in Table 12, an array of performance evaluation criteria, encompassing precision, re-call, sensitivity, specificity, F1-Score, accuracy, and others, were computed and documented in Table 13. Upon comparing the performance of the proposed approach with existing research in the realm of social media-based cyber intelligence, it be-comes apparent, as indicated in Table 14, that a few extant studies, specifically [17] and [21], outperformed the proposed approach in certain instances. Nonetheless, it is worth noting that the proposed approach exhibits superior performance compared to the majority of existing solutions documented in the literature. On average, the F1-Score achieved by the prevailing methodologies was observed to be 0.83, whereas the proposed solution showcased a significantly higher F1-Score of 0.88.
  • Secondly, the proposed approach disregarded the possibility that these Tweets could have originated from counterfeit accounts [39], or that genuine Twitter users may disseminate false information [40].
  • Thirdly, the study relies on real-time Tweet API, Microsoft Power Platform, and Microsoft Azure, all of which necessitate regular payment through credit cards. For in-stance, access to the basic Tweeter API with a monthly limit of reading only 10K Tweets incurs a cost of $100 USD per month [41]. Increasing this limit to read 1 mil-lion Tweets could result in a financial commitment of $5000 USD per month [41]. Consequently, in order to minimize expenses, this research examined only a limited number of Tweets. Researchers interested in working with real-time Tweets must possess access to credit cards and sufficient research funds to sustain the ongoing subscription costs.
  • Fourthly, this research extensively employed "black box" cloud-based services and tools, such as Microsoft Cognitive Services, which poses substantial challenges in investigating algorithmic biases and potential enhancements.
  • Lastly, this investigation employed industry-standard tools and cutting-edge cloud services, including Microsoft Power Platform and Microsoft Azure. Therefore, conducting this research necessitates expertise and certifications on these technologies and standards.

In future, our research endeavors will focus on the integration of advanced algorithms designed to identify and discern fake users, as well as counterfeit information, thereby facilitating the acquisition of more robust and validated cyber intelligence. Furthermore, we shall dedicate our efforts to further exploration of open-source tools and algorithms, aiming to mitigate cost obligations, enhance performance metrics (specifically in terms of F1-Score), and foster a greater sense of technological autonomy.”

Comment / Suggestion 3: Table 3 lacks a definition of notation (x): It is crucial to provide clear definitions of any notations used in the table for readers' understanding. This will help in comprehending the presented data accurately.

I concur with this valuable observation. Accordingly, I have added the definition of notation (X). Table 3 caption, now clearly states that “X denotes Supported”.

Comment / Suggestion 4: Lack of comparison of the proposed method's results with other reference methods from Table 3: To evaluate the effectiveness of the proposed solution, it is essential to compare its performance with existing reference methods. This comparison will provide a benchmark and help in assessing the superiority or effectiveness of the proposed approach.

I absolutely and wholeheartedly agree with this valuable point. Therefore, I have now added three new Tables (i.e., Table 12, Table 13, and Table 14) to evaluate the effectiveness of the proposed solution. Moreover, as per the suggestion of the honorable reviewer, I have now compared the performance of the proposed solution with existing methods (that appeared in Table 3). These newly added tables (i.e., Table 12, Table 13, and Table 14) will provide a benchmark and help in assessing the superiority or effectiveness of the proposed approach.

Round 2

Reviewer 1 Report

Thank you for addressing my feedback and enhancing the final manuscript. I believe that the provided explanations and additions have satisfactorily incorporated the requested changes. The manuscript now stands in a much improved position.

Reviewer 2 Report

Accept with current revision paper

 Minor editing of English language required